# Human kinesin-5 KIF11 drives the helical motion of anti-parallel and parallel microtubules around each other

Laura Meißner [1,4], Lukas Niese[1], Irene Schüring[1], Aniruddha Mitra[1,5] & Stefan Diez [1,2,3]✉

## Abstract

During mitosis, motor proteins and microtubule-associated protein organize the spindle apparatus by cross-linking and sliding microtubules. Kinesin-5 plays a vital role in spindle formation and maintenance, potentially inducing twist in the spindle fibers. The off-axis power stroke of kinesin-5 could generate this twist, but its implications in microtubule organization remain unclear. Here, we investigate 3D microtubule-microtubule sliding mediated by the human kinesin-5, KIF11, and found that the motor caused right-handed helical motion of anti-parallel microtubules around each other. The sidestepping ratio increased with reduced ATP concentration, indicating that forward and sideways stepping of the motor are not strictly coupled. Further, the microtubule-microtubule distance (motor extension) during sliding decreased with increasing sliding velocity. Intriguingly, parallel microtubules cross-linked by KIF11 orbited without forward motion, with nearly full motor extension. Altering the length of the neck linker increased the forward velocity and pitch of microtubules in anti-parallel overlaps. Taken together, we suggest that helical motion and orbiting of microtubules, driven by KIF11, contributes to flexible and context-dependent filament organization, as well as torque regulation within the mitotic spindle.

**Keywords** Helical Motion; Kinesin; Spindle; Torque
**Subject Categories** Cell Adhesion, Polarity & Cytoskeleton; Structural Biology

## Introduction

During mitosis, the spindle segregates the chromosomes to the emerging daughter cells. Reliable chromosome segregation is essential to cell survival, as segregation errors may result in chromosome instability and subsequently aneuploidy—a hallmark of several types of cancer. The spindle self-assembles into a metastable structure with microtubules as basic building blocks. Kinesin and dynein motors organize the microtubules into spindle fibers by cross-linking and sliding them (Walczak et al, 1997; Sharp et al, 2000). One of the essential motors in spindle organization is kinesin-5. In prophase, kinesin-5 generates extensile pushing forces that slide anti-parallel microtubules apart for the segregation of the duplicated centrosomes (Blangy et al, 1995; Sharp et al, 1999). During metaphase and anaphase, the sliding activity of kinesin-5 contributes to spindle elongation and its cross-linking activity stabilizes microtubule bundles (Heck et al, 1993; Brust-Mascher et al, 2009). Kinesin-5 additionally localizes to the spindle poles, where it bundles and focuses parallel microtubules (Sawin et al, 1992; Mann and Wadsworth, 2018). Inhibition of kinesin-5 impairs pole separation, resulting in monopolar spindles in *Xenopus laevis*, monkey, and human cells (Walczak et al, 1998; Mayer et al, 1999; Kapoor et al, 2000). Kinesin-5 inhibition after spindle assembly causes spindle defects in anaphase B and telophase with effects on positioning of the daughter nuclei in *Drosophila melanogaster* (Sharp et al, 1999). Thus, kinesin-5 is indispensable for the correct functioning of mitosis.

The mechanisms of microtubule organization by kinesin-5 have so far been described by structural and two-dimensional (2D) in vitro experimental studies. Kinesin-5 is a bipolar tetramer, with two motor domains on each side, located at the N-terminus (Kashina et al, 1996). Each pair of motor domains binds one microtubule, in this way cross-linking the filaments. Upon motor stepping, the microtubules slide apart when their orientation is anti-parallel (microtubule polarity in opposite direction), whereas parallel microtubules do not slide (Kapitein et al, 2005). In addition to forward motion, kinesin-5 displays a sideways stepping component, which results in an orthogonal, sideways motion. When a truncated, dimeric construct of KIF11 was immobilized on a surface, it propelled microtubules forward and simultaneously rotated them in a left-handed manner (Yajima et al, 2008).

To explore if sideways components in the power strokes of cross-linking motors can induce helical motion of sliding microtubules around each other, we have recently developed a three-dimensional (3D) motility assay, in which microtubules are

[1]B CUBE – Center for Molecular Bioengineering, TUD Dresden University of Technology, 01307 Dresden, Germany. [2]Max Planck Institute for Molecular Cell Biology and Genetics, 01307 Dresden, Germany. [3]Cluster of Excellence Physics of Life, TUD Dresden University of Technology, 01062 Dresden, Germany. [4]Present address: BASS Center, Molecular Biophysics and Biochemistry Department, Yale University, 06511 New Haven, USA. [5]Present address: Cell Biology, Neurobiology and Biophysics, Department of Biology, Faculty of Science, Utrecht University, 3584CH Utrecht, Netherlands. ✉E-mail: stefan.diez@tu-dresden.de

suspended on micro-structured polymer ridges (Mitra et al, 2018; Bugiel et al, 2018). In this assay, the *Drosophila melanogaster* kinesin-14, Ncd, has been shown to drive the right-handed helical motion of short cargo microtubules around the freely suspended sections of immobilized ('fixed') microtubules (Mitra et al, 2020). Kinesin-5 is anticipated to show a similar behavior but may display different biophysical and structural properties because of its distinct role in the spindle compared to kinesin-14.

In vivo, it has been proposed that the 3D motility of kinesin-5 influences the shape of the spindle, as spindle fibers deform under force. Recently, high-resolution imaging of HeLa cells revealed, that the spindle is twisted into a chiral structure. Inhibition of kinesin-5 abolished the twist, whereas overexpression did not change it (Novak et al, 2018; Trupinić et al, 2022). This implies that the sideways stepping component of kinesin-5 might be involved in generating rotational forces (torques), which twists the spindle with a specific chirality. In contrast to HeLa cells, spindles of RPE1 cells did not exhibit helicity or a weaker twist and became strongly twisted upon double knockout of kinesin-5 and the dynein targeting factor NuMA, which rendered the spindles fragile (Trupinić et al, 2022; Neahring et al, 2021). Thus, the role of twist is not yet fully understood and, though being a key determinant for spindle shape, the 3D motility behavior of kinesin-5 remains elusive.

Here, we show that human kinesin-5, KIF11, drives the helical motion of anti-parallel microtubules around each other. Interestingly, a similar orbiting motion is also observed for parallel microtubules, though those microtubules are not moving in forward direction. Variation of the critical mechanical element of the motor, the neck linker, increased the forward velocity and helical pitch of KIF11. This suggests, that KIF11 constitutes a slowly sliding, fast rotating motor—exhibiting different motility modes dependent on the microtubule orientation.

## Results

To study the 3D motility of microtubules driven by KIF11, we performed 3D sliding assays on micro-structured polymer ridges, with 10 μm wide valleys, separated by 360 nm high and 2 or 5 μm wide ridges (Fig. 1A, Methods). The micro-structures were coated with TAMRA antibodies to suspend long, TAMRA-labeled, 'fixed' microtubules. By bridging from ridge to ridge, the lattice of the fixed microtubule is freely accessible over the valley regions. Subsequently, KIF11 with a C-terminal EGFP tag (referred to as KIF11 throughout this work, Appendix Fig. S1, Methods, 10 nM) and 1–4 μm long, Atto647N labeled 'cargo microtubules' were added in an ADP containing buffer. The sliding process was initiated by adding ATP. Cross-linked cargo and fixed microtubules were tracked using the MATLAB-based software FIESTA (Ruhnow et al, 2011) and the perpendicular distance of the center point of the cargo microtubule to the averaged position of the fixed microtubule was measured (referred to as sideways distance). Per our definition, negative sideways distances correspond to movement of the cargo microtubule on the right side of the fixed microtubule (when viewed from the trailing end of the cargo microtubule in the images, Methods).

To analyze the 3D motion, the sideways distance of sliding cargo microtubules was plotted with respect to the traveled forward distance (Fig. 1B; Movie EV1). For the given example, the sideways distance undulated between −100 and 100 nm (showing five full periods) over the valley region, which is reminiscent of a helical motion. Because helical motion was impaired on the ridges, the cargo microtubule was pressed to one side of the fixed microtubule during the first and last 2 μm of forward motion, resulting in a constant sideways distance of around −70 nm. Considering the optical setup of the utilized inverted fluorescence microscope, negative sideways distances on the ridges correspond to a right-handed helical motion (Methods). To confirm the handedness of the helical motion, we lowered the focal plane to the height of the valleys; thus, microtubules were located above the focal plane and the fluorescence intensity increased when the cargo microtubule went from the top of the fixed microtubule to the bottom (Methods). A maximum of the fluorescence intensity was then followed by a maximum of the sideways distance with a phase shift of π/2, which confirmed the right-handedness of the helical motion for 12 cargo microtubules (Fig. 1C). In total 88 cargo microtubules displayed similar, robust helical motion over the valley regions (Fig. 1D, breakdown of datasets in Appendix Tab. S1).

The trajectories of the cargo microtubules contain information about their motility parameters. For analysis, only microtubules with at least two full rotations were considered. The motility parameters were calculated for each rotation and averaged for each cargo microtubule (Methods). The forward velocity was 27.0 ± 2.6 nm/s (mean ± standard deviation, $n = 88$ cargo microtubules, Fig. 1E), the angular velocity 0.114 ± 0.023 rad/s (Fig. 1F), and the pitch 1.54 ± 0.31 μm (Fig. 1G). Control experiments using KIF11 without EGFP revealed a similar helical motion of cargo microtubules (Appendix Fig. S2A,B). Neither forward velocity, nor angular velocity, nor pitch correlated with the lengths of the cargo microtubules (Fig. 1E–G; Appendix Fig. S3A–C, Pearson coefficients <0.3). We therefore reasoned that the rather large spreads in the motility parameters (in particular angular velocity and pitch) were not due to the length distribution of the cargo microtubules. To investigate if the variability in the motility parameters is inherent or is influenced by variations in the lattice structures of the fixed microtubules, we analyzed eight fixed microtubules, which were traversed by two to six cargo microtubules each. We found that the standard deviations of forward velocity, angular velocity, and pitch obtained from cargo microtubules sliding on the same fixed microtubule were similar or higher than the standard deviations of all cargo microtubules on all fixed microtubules: maximum values of 16.6% versus 20.1%, 26.2% versus 9.6%, 30.4% versus 20.2%, respectively (Appendix Fig. S3D versus Fig. 1E–G). This implies that the inherent variability in the motility parameters is not dependent on the variability in the structure of the fixed microtubules.

Another parameter, which could influence the motility parameters, is the motor density in the overlaps. Over time, motors might detach from the microtubules, leading to a decrease in their density. However, we did not observe a pronounced change in forward velocity over time. In addition, we tested various motor concentrations of KIF11 (2 nM, 5 nM, and 50 nM) in 3D sliding assays (Appendix Fig. S4A–D). Under almost all conditions the means of the motility parameters were in the same range. Likewise, the buffer composition (BRB80 without Tween-20, BRB40 and 20 mM Hepes with 50 mM KCl, i.e., with similar ionic strength to BRB80) did not have a strong effect on the motility parameters

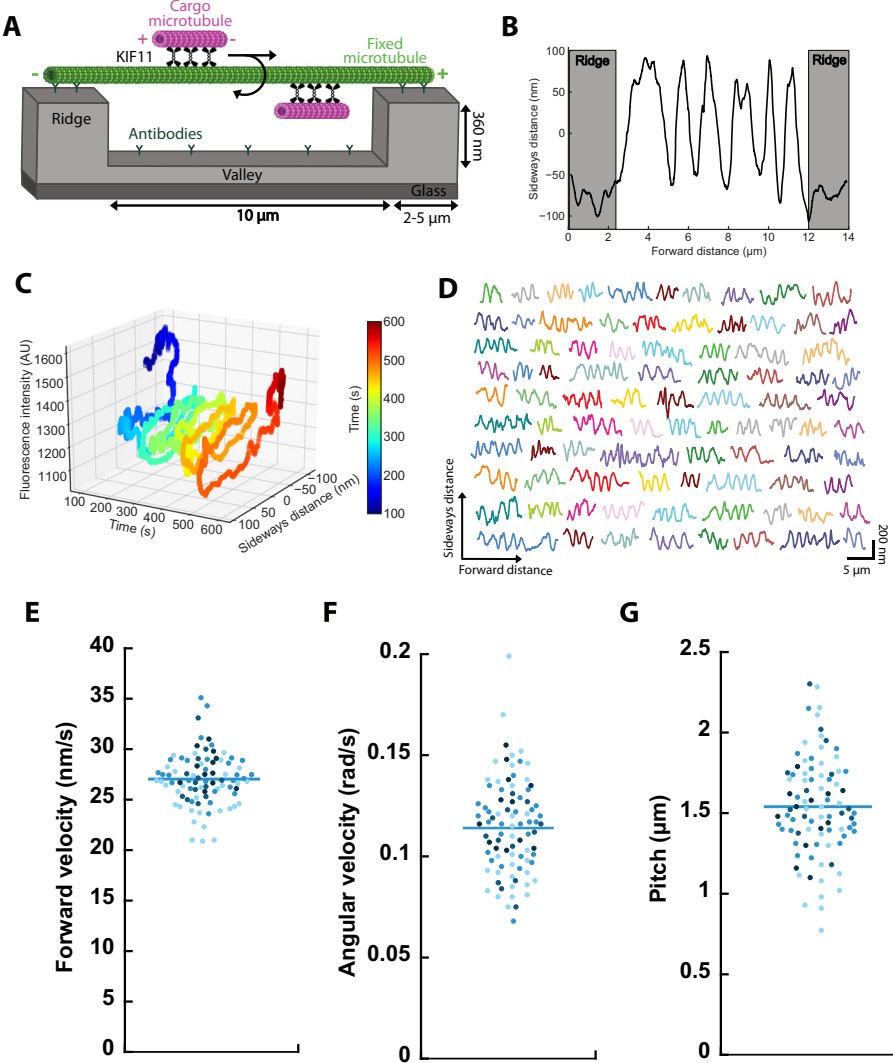

**Figure 1.  KIF11 drives the right-handed helical motion of anti-parallel cargo microtubules around fixed microtubules.**

(A) Setup of the 3D sliding motility assay. Fixed microtubules were suspended on ridge micro-structures, allowing a helical motion of the cargo microtubules driven by KIF11 over the valley regions. (B) Sideways distance as function of forward distance for an exemplary cargo microtubule. (C) 3D analysis of cargo microtubule motion. The phase-shifted, periodic changes in the fluorescence intensity and sideways distance of the cargo microtubule are consistent with a right-handed helical motion. (D) Example tracks of cargo microtubules driven by KIF11. (E–G) Forward velocity, angular velocity, and pitch of the example cargo microtubules from (D). Color coding indicates cargo microtubule length: <1.5 µm (light blue), <2.2 µm (sky blue), <2.9 µm (medium blue), >=2.9 µm (dark blue). Bars indicate mean values (D–G: $n = 88$ cargo microtubules). Source data are available online for this figure.

(Appendix Fig. S5A–D). The Hepes-based buffer slightly increased most motility parameters, but no difference between BRB80 without Tween-20 and BRB40 was observed.

To test if the sideways motion of cargo microtubules is strictly coupled to their forward motion, we varied the forward velocity by applying different ATP concentrations (1 mM to 25 µM; 1 mM: $n = 88$ cargo microtubules, 250 µM: $n = 4$, 200 µM: $n = 35$, 150 µM: $n = 24$, 30 µM: $n = 8$, 25 µM: $n = 8$). The forward velocity decreased from $27.0 \pm 2.6$ nm/s to $5.8 \pm 0.7$ nm/s and the angular velocity from $0.114 \pm 0.023$ rad/s to $0.057 \pm 0.004$ rad/s (Fig. 2A). Thereby, the angular velocity showed a weak linear correlation with the forward velocity below 20 nm/s but saturated above 20 nm/s. The

pitch decreased with decreasing forward velocity from $1.54 \pm 0.31$ µm to $0.66 \pm 0.09$ µm (Pearson correlation coefficient 0.84, Fig. 2B). The sidestepping ratio, calculated as the ratio of sideways movement (in protofilament steps, i.e., in units of $2\pi/14$) to forward displacement (in steps of 8 nm) decreased with increasing velocity from 0.29 to 0.06 (Pearson correlation coefficient $-0.86$, Fig. 2C). This indicates, that the slower a cargo microtubule moved in the longitudinal direction, the more likely it moved sideways in the axial direction. In addition, we segmented the tracks into ridge and valley parts and grouped them into two types of transitions: from ridge to valley and from valley to ridge. To test for changes in forward velocity at these transitions, we

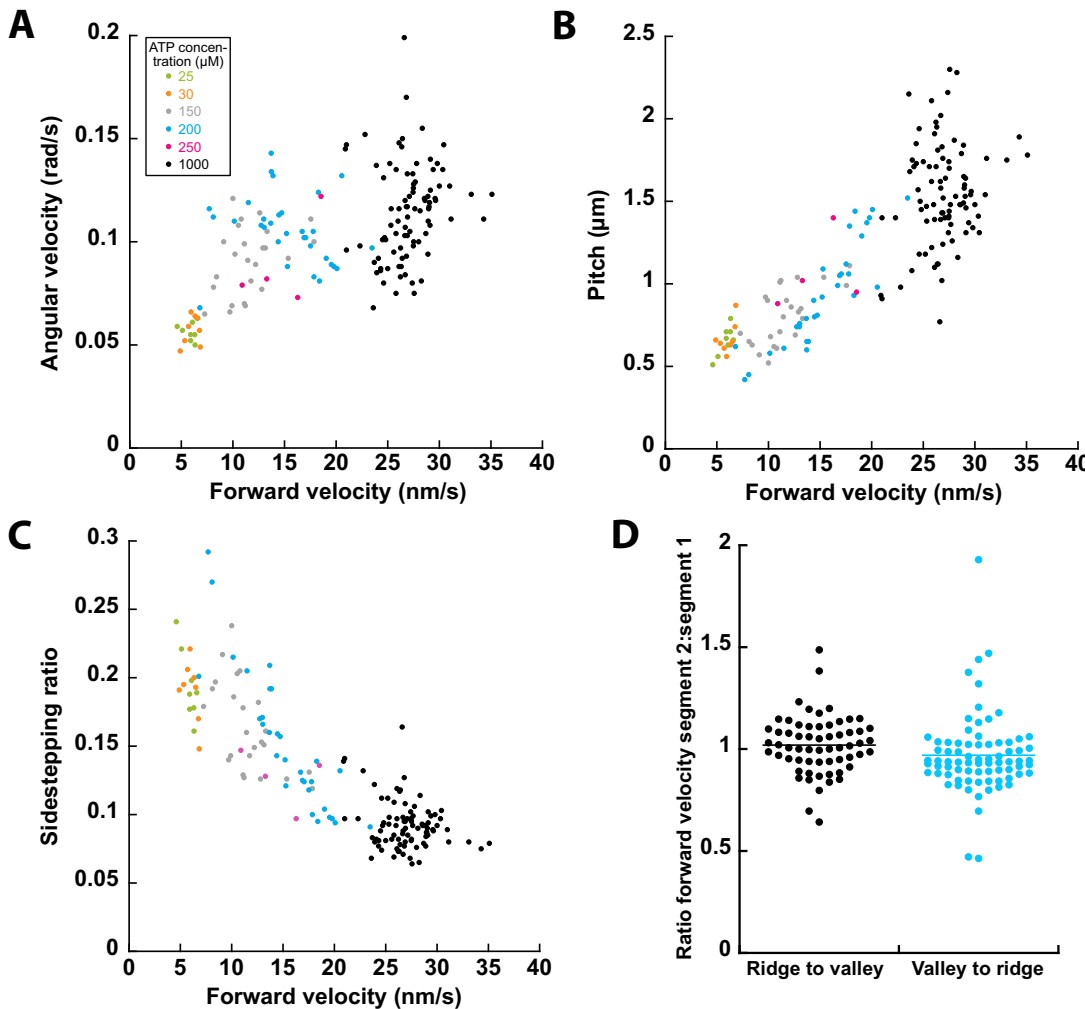

**Figure 2. Sideways and forward motion of cargo microtubules driven by KIF11 are not strictly coupled.**

(A) Angular velocity, (B) pitch, and (C) sidestepping ratio correlated with forward velocity. 1 mM: $n = 88$ cargo microtubules, 250 µM: $n = 4$, 200 µM: $n = 35$, 150 µM: $n = 24$, 30 µM: $n = 8$, 25 µM: $n = 8$. (D) The forward velocity of cargo microtubules did neither change significantly at transitions from ridge to valley nor at transitions from valley to ridge. Ridge to valley: $n = 61$ cargo microtubules, valley to ridge: $n = 76$. Source data are available online for this figure.

calculated the velocity ratios of both segments (segment 2 divided by segment 1, Fig. 2D, ridge to valley: $n = 61$ cargo microtubules, valley to ridge: $n = 76$). We observed that the cargo microtubules did not change their forward velocity at ridge-valley transitions, indicating that suppression of the sideways motion did not affect forward motion. Taken together, these findings show that sideways and forward motion are not strictly coupled.

In our 3D sliding assays, not all cargo microtubules performed a helical motion around the fixed microtubules over valleys. Besides the canonical helical motion (i.e., forward and sideways, Figs. 1 and 2), some cargo microtubules only moved forward (33.7%, forward-only cargo microtubules), while others neither moved forward nor sideways (27.8%, stuck cargo microtubules). For these cargo microtubules we cannot rule out that a helical motion was not detected due to technical reasons (Methods). However, 24.9% of the cargo microtubules did not move forward significantly (forward velocity slower than 5 nm/s) but showed robust orbiting around the fixed microtubules (Fig. 3A, sideways-only cargo microtubules, Appendix Fig. S6A–D; Movie EV2).

Generally, while a cargo microtubule oriented anti-parallel to a fixed microtubule is expected to move forward, parallel microtubules are locked in the longitudinal direction (Fig. 3B (Kapitein et al, 2005)). Hence, we conjectured that the helically moving cargo microtubules were anti-parallel to the fixed microtubules, while the sideways-only cargo microtubules were parallel. To test this hypothesis, we employed polarity-labeled cargo microtubules and used the pronounced residence time and accumulation of KIF11-EGFP on the plus ends of the fixed microtubules as markers for the polarity of the fixed microtubules (Methods). We detected five events of helically moving cargo microtubules with four of them in an anti-parallel and one in a parallel orientation (Appendix Fig. S6E; Movie EV3). In contrast, from 18 sideways-only cargo microtubules all of them were in a parallel configuration (Appendix Fig. S6F; Movie EV4), confirming our hypothesis. In addition, we observed a number of events where a cargo microtubule did not move forward initially but began to move forward rapidly, after flipping its orientation, with helical motion in anti-parallel orientation and orbiting in parallel orientation. Sideways-only

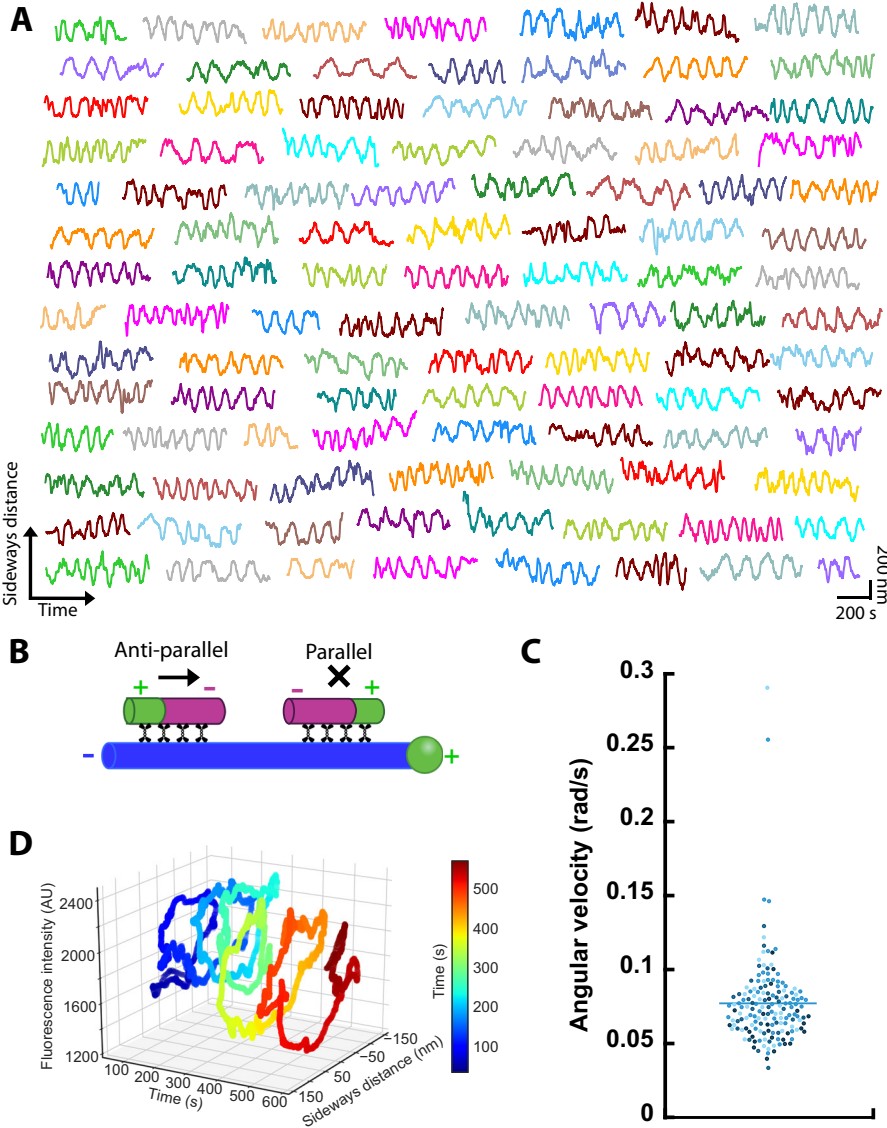

**Figure 3. KIF11 orbits parallel cargo microtubules around fixed microtubules without significant forward movement.**

(A) Example tracks of cargo microtubules driven by KIF11 ($n = 103$ cargo microtubules). (B) Schematics of microtubule orientation-dependent motility modes of KIF11 in forward direction. (C) Angular velocities of cargo microtubules moving sideways-only. Color coding indicates cargo microtubule length: <1.5 μm (light blue), <2.2 μm (sky blue), <2.9 μm (medium blue), >=2.9 μm (dark blue). Bar indicates mean value ($n = 161$ cargo microtubules). (D) 3D analysis of cargo microtubule motion. The phase-shifted, periodic changes in the fluorescence intensity and sideways distance of the cargo microtubule are consistent with a right-handed orbiting when viewed from the minus end of the cargo microtubule. Source data are available online for this figure.

cargo microtubules exhibited a 1.5-fold lower angular velocity ($0.077 \pm 0.030$ rad/s) than the forward and sideways moving cargo microtubules (Fig. 3C, compared to Fig. 1F). Again, no correlation with the lengths of the cargo microtubules was observed. In the control experiments with a lowered focal plane, three cargo microtubules orbited in a right-handed manner when viewed from the minus end of the fixed microtubule (Methods). Taken together, KIF11 drives microtubule-microtubule sliding in at least two modes: (i) cargo microtubules which are anti-parallel to a fixed microtubule move forward in a right-handed helical manner around the fixed microtubule with fast angular velocity and (ii) cargo microtubules, which are parallel to the fixed microtubule, orbit in a right-handed

manner around the fixed microtubule with slower angular velocity, without forward movement (Fig. 3D).

An additional parameter that can be estimated from our measurements is the distance between cargo and fixed microtubules during helical motion or orbiting. Under the assumption that the motors bind both microtubules at their shortest distances (i.e., on the protofilaments facing each other), this value represents the extension of the active motors perpendicular to the microtubules (referred hereafter as 'motor extension'). To this end, we measured the distance between the minima and maxima of the sideways distance and determined the motor extension, as illustrated in Fig. 4A and described in Methods). At 1 mM ATP concentration we

# A

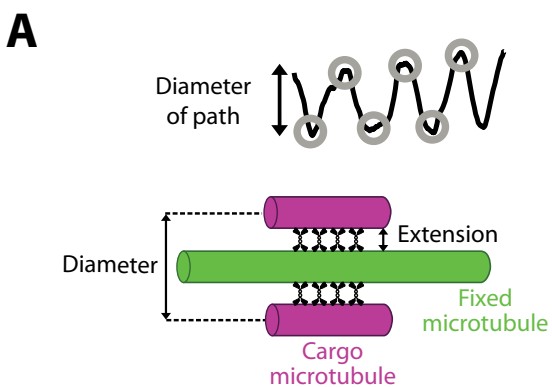

# B

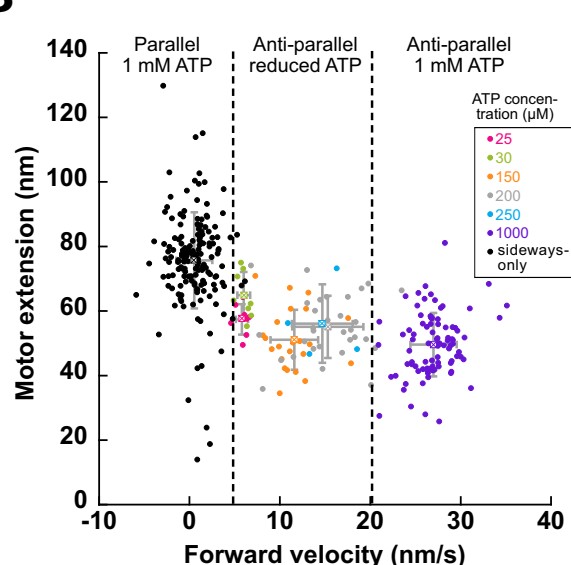

**Figure 4. KIF11 motor extension in microtubule overlaps depends on microtubule orientation and forward velocity.**

(A) Schematics to estimate the motor extension. (B) The motor extension of KIF11 correlated with forward velocity (with the lowest extension at highest velocity). Motors in sideways-only events displayed the highest extension (25 μM: *n* = 8 cargo microtubules, 30 μM: *n* = 8, 150 μM: *n* = 24, 200 μM: *n* = 35, 250 μM: *n* = 4, 1 mM anti-parallel: *n* = 88, 1 mM parallel sideways only: *n* = 161, mean with standard deviation are shown for each condition). Source data are available online for this figure.

three residues in the same protein changed its rotational pitch (from 8.7 μm of the supertwist to 4.4 μm (Mitra et al, 2018). Contrary to this, the velocity and force of kinesin-5/kinesin-1 chimeras were independent of the neck linker length (Düselder et al, 2012). We either inserted glycine and serine residues at the C-terminus (GS, NL20 and GSGS, NL22) or removed the C-terminal two or four residues (NL16 and NL14) and tested all constructs in 3D sliding motility assays. The change of the neck linker length did not affect the mobility of cargo microtubules. For the wild type, 47.3% of cargo microtubules moved forward (*n* = 306) and 52.7% were immobile in forward direction (*n* = 341, breakdown of datasets Appendix Tab. S1). The neck linker mutants displayed a similar fraction of mobile cargo microtubules, between 36 and 58% (all mutants pooled: *n* = 660). The mean forward velocity drastically increased for all constructs, from 27 ± 3 nm/s of the wild type to 74 ± 29 nm/s (*n* = 12), 48 ± 20 nm/s (*n* = 11), 127 ± 34 nm/s (*n* = 18), and 136 ± 15 nm/s (*n* = 7) for NL14, NL16, NL20, and NL22, respectively (Fig. 5B). Similarly, the angular velocity of the neck linker mutants doubled to tripled compared to the wild type from 0.11 to 0.23–0.30 rad/s (Fig. 5C). The pitch of NL16 (1.2 ± 0.5 μm) was similar to NL18-WT (1.5 ± 0.3 μm) and the pitch of NL14 increased to 2.4 ± 1.5 μm, whereas the pitch of the elongated neck linkers more than doubled to 3.5 ± 2.0 (NL20) and 4.2 ± 1.5 μm (NL22, Fig. 5D). Notably, the distribution of both forward velocity and pitch broadened for all neck linker mutants. As previously observed for NL18-WT, the pitch of the neck linker mutants increased with the forward velocity, showing low pitches at low forward velocities (Fig. 5E). Similar to NL18-WT, we observed a significant fraction of orbiting cargo microtubules (NL14: one event, NL16: five events, NL20: 15 events, NL22: one event). Thus, the neck linker mutants retain the different motility modes but display drastically altered and more variable motility parameters.

## Discussion

Previously, kinesin-5-driven microtubule-microtubule sliding has been studied in 2D motility assays with fixed microtubules fully immobilized on glass surfaces. Such assays have been limited to observing the forward motion of cargo microtubules. To investigate the sideways stepping component of motors, 3D setups are required, as demonstrated in recent studies (Brunnbauer et al, 2012; Bugiel et al, 2018; Can et al, 2014; Mitra et al, 2018). Using a 3D setup, where fixed microtubules were elevated on micro-structured polymer ridges, we showed that KIF11 drives the helical motion of cargo microtubules around fixed microtubules. This is the second report of such helical motion for microtubule cross-linking motors after an earlier demonstration for kinesin-14, Ncd (Mitra et al, 2020) and the first report for a plus-end directed motor.

Our observations indicate a right-handed helical motion of KIF11-driven cargo microtubules. We confirmed the direction additionally by direct comparison of the sign of the sideways distance to 3D sliding assays with Ncd. Previous reports found that plus-end directed motors (e.g., kinesin-6 and kinesin-8) move along microtubules exclusively in a left-handed manner, whereas minus-end directed motors (such as kinesin-14 and cytoplasmic dynein) move in a right-handed manner (Maruyama et al, 2021; Mitra et al, 2018; Nitzsche et al, 2016; Can et al, 2014; Mitra et al, 2020; Walker

determined a motor extension of 49.6 ± 9.8 nm (*n* = 88 overlaps, Fig. 4B) which is about 60% of the motor contour length of 79 nm obtained from electron microscopy (Scholey et al, 2014). Upon lowering the velocities by reducing the ATP concentration, the motor extension increased to 64.9 ± 7.2 nm (*n* = 8). The motor extension in the sideways-only events yielded even higher values of 75.7 ± 14.9 nm (*n* = 161), close to the motor contour length. Thus, the motors adopt a larger extension lower the forward velocity and reach almost the maximum extension in absence of forward motion.

Finally, we sought to perturb the sidestepping of KIF11 by varying the length of the neck linker domain to identify how it contributes to the sidestepping (NL18-WT, Fig. 5A). Previous work found that neck linker length and rigidity affected the sidestepping rate of *Drosophila melanogaster* kinesin-1, KHC. There, an insert of

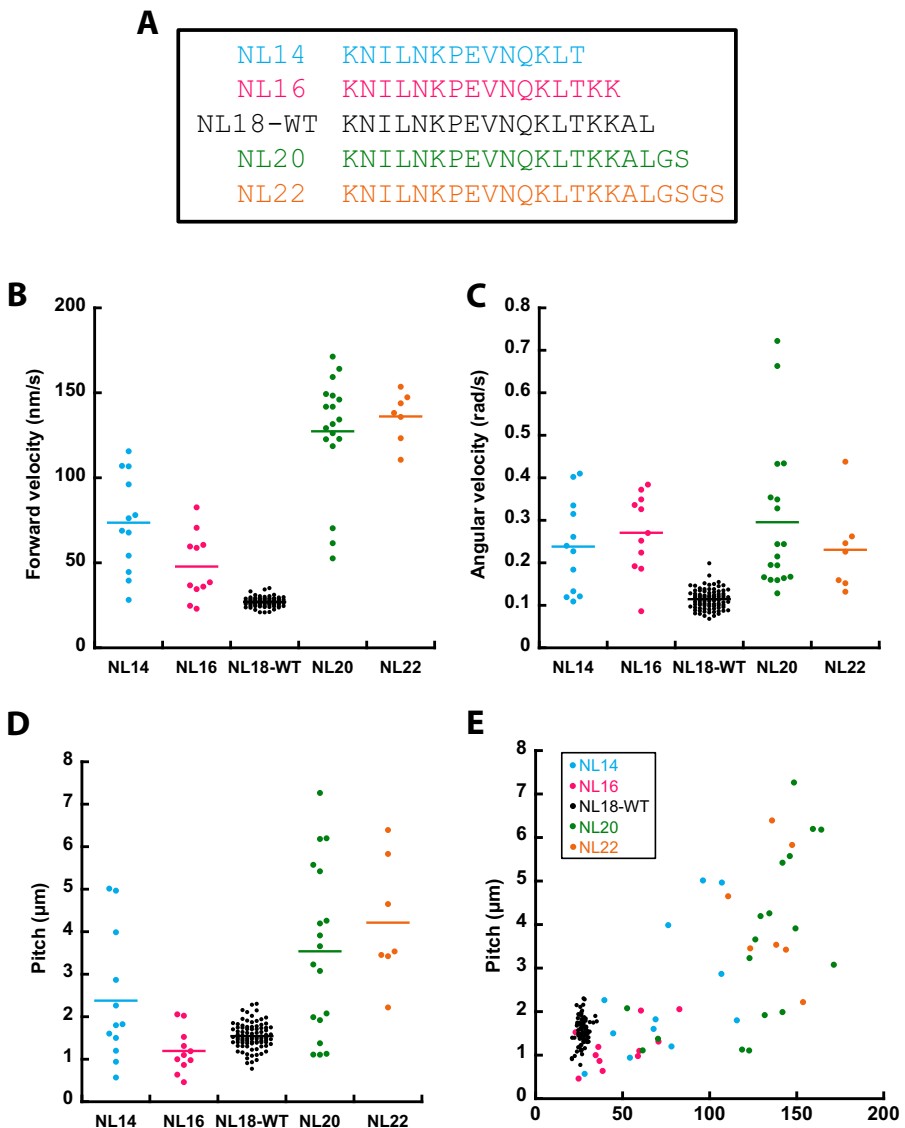

**Figure 5. Changing the neck linker length of KIF11 affects the motility parameters.**

(A) Overview of neck linker mutant constructs. (B) Forward velocity, (C) angular velocity, and (D) pitch of cargo microtubules driven by the neck linker mutants in comparison to wild type. (E) The pitch showed a correlation with forward velocity. NL14: $n = 12$ cargo microtubules, NL16: $n = 11$, NL18-WT: $n = 88$, NL20: $n = 18$, NL22: $n = 7$. Source data are available online for this figure.

et al, 1990). We note that the handedness is identical in different assay geometries: left-handed stepping of the motors on microtubules corresponds to a left-handed rotation of microtubules around their long axis in gliding assays and a left-handed helical motion of cargo microtubules around fixed microtubules in 3D sliding assays. The right-handedness in our experiments is also contradictory to previous observations of single-headed human Eg5 (KIF11) and truncated (528 N-terminal residues) yeast kinesin-5 Cin8, which both rotate microtubules in gliding assays in a left-handed manner (Yajima et al, 2008; Yamagishi et al, 2021). It is possible that these conflicting findings arise from differences in protein structure, as KIF11 and Cin8 only share 45% sequence identity (528 N-terminal residues). In addition, truncation of the

motor can affect its motility, as shown for kinesin-1, which displayed a torque component only in the truncated, monomeric form (Yajima and Cross, 2005).

The mean pitch of the helical motion of $1.5 \pm 0.3$ µm in our experiments cannot be related to the supertwist of GMP-CPP grown microtubules, which mainly consist of 14 protofilaments, resulting in a left-handed supertwist of about 8 µm (Hyman et al, 1995; Nitzsche et al, 2008; Ray, 1993). Thus, helical motion must arise from a sideways stepping component of KIF11. The pitch is similar to the helical motion driven by Ncd (median pitch of 1.6 µm, (Mitra et al, 2020)) and dimeric human Eg5 (KIF11) (pitch of 2.3 µm, (Yajima et al, 2008)), but five times higher than the pitch of single-headed KIF11 and truncated Cin8 in surface gliding assays

(pitches of about 0.3 μm (Yajima et al, 2008; Yamagishi et al, 2021)). We believe, the latter difference might be attributed to the different motor constructs and different motor properties for different organisms rather than to the assay geometry.

In our experiments, the angular velocity and the pitch of cargo microtubules displayed large spreads of 0.07–0.20 rad/s and 0.8–2.3 μm, respectively. We could neither attribute these spreads to cargo microtubule length nor forward velocity. We also ruled out that angular velocity and pitch are set by the fixed microtubule, because different cargo microtubules on the same fixed microtubule moved with highly variable motility parameters. Moreover, different motor concentrations displayed similar motility parameters. Nevertheless, local inhomogeneities in motor density (potentially involving motor accumulation at the plus end of the cargo microtubule) could play a role as indicated by the higher variability of the motility parameters for shorter cargo microtubules (Appendix Fig. S3).

Our observation of similar velocities over ridges and valleys suggests that unimpeded forward motion is possible when sideways motion is suppressed, suggesting that the two motions are not strictly coupled. To validate this hypothesis, we conducted experiments in which we reduced the ATP concentration from 1 mM to 25 μM, resulting in a decrease in forward velocities to 5.8 ± 0.7 nm/s. The sidestepping ratio then more than doubled, rising from 0.09 ± 0.02 at 1 mM ATP to 0.19 ± 0.03 at 25 μM ATP. These results indicate that at lower ATP concentrations and subsequently lower forward velocities, KIF11 is more inclined to step sideways. This finding further supports the notion that forward and sideways motion are not strictly coupled, as a strict coupling would have resulted in a constant sidestepping ratio regardless of the forward velocity.

Similar to Kip3, KIF11 is a processive motor which exhibits a dependence of rotations on ATP concentration. For Kip3, a model was proposed where the motor spends an extended period in the ATP waiting state (Mitra et al, 2018). During this waiting time, the motor has a stochastic probability to switch from a less-diffusive two-head bound state to a more diffusive one-head bound state. Upon ATP binding, the motor is significantly more likely to sidestep if it is in the one-head bound state. Thus, the longer the motor waits for ATP, the more likely it is to switch to a one-head bound state and sidestep. It is possible that a similar stepping model, incorporating both ATP-dependent and ATP-independent components, may apply to KIF11.

Besides the helical motion of anti-parallel microtubules, we observed that KIF11 has at least one other motility mode, in which cargo microtubules orbit around fixed microtubules without forward movement. Polarity-labeling of the microtubules revealed for this "sideways only" motility mode a parallel orientation of the microtubules. The lack of forward motion for parallel microtubules is not surprising because the forward movement of motors is expected to cancel out in parallel microtubules (Fig. 6A, black and gray straight arrows, (Kapitein et al, 2005; Fink et al, 2009)). Helical motion and orbiting, however, arise from the sideways stepping of the motor domains on the fixed microtubule, independent of the orientation of the cargo microtubule (Fig. 6A, black and blue circular arrows). Ncd did not drive the orbiting of cargo microtubules around fixed microtubules in parallel arrangement, although the same geometrical considerations should apply to this motor (Mitra et al, 2020). We can only speculate that the non-

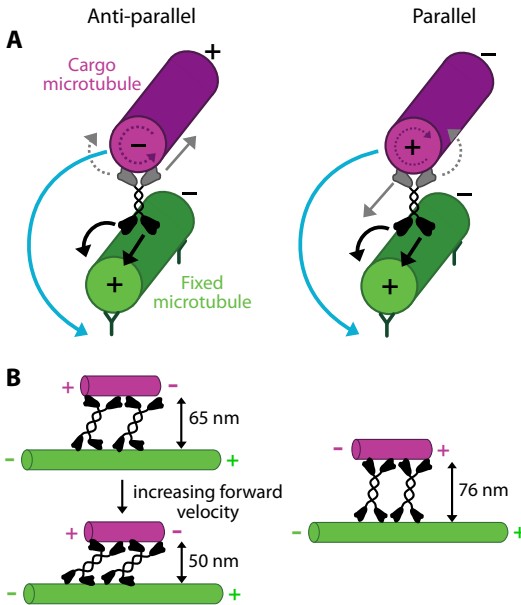

**Figure 6.** Schematics for helical sliding of anti-parallel microtubules and orbiting for parallel microtubules. **(A)** Forward and sideways stepping directions of KIF11 in anti-parallel and parallel microtubule overlaps. Rotational motion occurs in both cases. Solid arrows indicate observed motion, dotted arrows inferred motion. **(B)** The extension of KIF11 decreases for increasing sliding velocity of anti-parallel microtubules (left) and adopts an almost fully extended conformation (contour length of KIF11 about 79 nm) in parallel overlaps (right).

processive movement of individual Ncd motors prevented persistent orbiting. In addition to the helical or orbiting motion of the cargo microtubules around the fixed microtubules, we infer that the KIF11 motor domains rotate the cargo microtubules around their own axes, with different rotation directions for anti-parallel and parallel arrangements (Fig. 6A, gray and purple circular, dotted arrows). It will be intriguing to simultaneously monitor this rotation in addition to the helical and orbiting movement in future experiments.

Compared to the helical motion of anti-parallel cargo microtubules, the angular velocity of the orbiting cargo microtubules was lower by a factor of 1.5. This decrease could possibly be attributed to the conformation of the motor domain in the overlaps, which is set by the orientation of the tubulin dimers: The motor domains of KIF11 interacting with the two microtubules have to point in opposite directions in the anti-parallel case and in the same direction in the parallel case. On the other hand, structural data of KIF11 showed, that the pairs of motor domains on each side have an offset of 90°. Thus, the motor might have to twist more in the parallel case, which could result in an increased strain in the motor and therefore a less efficient stepping cycle.

At 1 mM ATP KIF11 adopted an extension of 50 nm perpendicular to the microtubules in anti-parallel overlaps. The extension increased to 65 nm with decreasing forward velocity (Fig. 6B, left panel) and was maximal with 76 nm in parallel microtubule overlaps, i.e., in absence of forward motion of the cargo microtubules (Fig. 6B, right panel). This suggests that the motors' native binding state, without load, is rather perpendicular to the microtubule leading to a large extension, which is decreased

during sliding, possibly due to the viscous drag experienced by the cargo microtubule. We hypothesize that motor extension might be another factor for motor functioning and regulation in vivo because other microtubule-associated proteins are present in microtubule overlaps. For example, the passive cross-linker PRC1 holds microtubules apart by about 35 nm (Subramanian et al, 2010) and kinesin-14 slides microtubules with an extension of about 20 nm (Mitra et al, 2020). As kinesin-5 and kinesin-14 are antagonists, it will be interesting to investigate (i) what is the distance between overlapping microtubules in the presence of both motors, (ii) if kinesin-5 geometrically alters the activity and binding kinetics of kinesin-14, because the latter may not be able to cross-link both microtubules in the presence of kinesin-5, and (iii) if PRC1 influences the extension of both motors during sliding.

We further explored how perturbation of the neck linker affects the 3D motion of KIF11. The neck linker, which connects the motor domain to the stalk, resembles the key mechanical element of the motor, because it transmits force from the motor domain to the stalk and modulates the microtubule-microtubule affinity (Khalil et al, 2008; Hwang et al, 2008). In the ADP state, the neck linker points towards the microtubule minus end. Upon ATP binding the neck linker docks and reorients itself towards the microtubule plus end. In this conformation, the neck linker forms the cover neck bundle with the N-terminal extension (Goulet and Moores, 2013). Previously, it has been shown that kinesins, which track a single protofilament, possess a shorter neck linker (kinesin-1: 14 residues) than sidestepping kinesins (kinesin-2 and kinesin-8: 17 residues, kinesin-5: 18 residues (Shastry and Hancock, 2010; Bormuth et al, 2012; Hariharan and Hancock, 2009)). Changing the neck linker can alter the motor's stepping in three ways: by disrupting the cover neck bundle, by affecting the motor affinity to the microtubule, and by influencing the geometry of binding. For the last case, the length of the neck linker could determine, how efficiently the lagging head reaches the next tubulin dimer with its step. Both longer and shorter neck linkers increased forward velocity (1.8 to 5.0-fold), angular velocity (doubling to tripling) and pitch (for three of the four constructs, 1.6- to 2.8-fold). The higher increase in forward than angular velocity resulted in a lower sidestepping ratio of the mutants compared to the wild type, which decreased the pitch of the mutants. Previous work showed that the deletion of the tail increased the forward velocity of KIF11, but came at the expense of less force production (Bodrug et al, 2020), which could be the case for the neck linker mutants as well. Thus, we speculate that KIF11 constitutes a slow, potentially high force, motor with a high sidestepping ratio.

In cells, parallel microtubules are found near the spindle poles, where they are rigidly anchored with their minus ends to the same microtubule-organizing center (MTOC). We speculate that KIF11 causes these microtubules to wrap around each other without generating supertwist of the resulting bundles. In contrast, anti-parallel microtubules occur in the spindle midzone and are anchored with their minus ends to opposite MTOCs. These microtubules are expected to roll against each other while sliding apart, which leads to a supertwist of the microtubule bundles. Right-handed helical motion in the midzone is expected to lead to a right-handed twist of the spindle fibers, which is the opposite of the observed left-handed twist of spindles in HeLa cells (Novak et al, 2018). As Trupinic et al demonstrated, the perturbation of other

motors including kinesin-6 MKLP1, kinesin-8 KIF18A, and cytoplasmic dynein, as well as the microtubule nucleation factor augmin and the passive microtubule cross-linker PRC1 affect the spindle twist in different ways (Trupinić et al, 2022). Thus, other factors besides kinesin-5 might contribute to the twist and other motors could be alternative torque generators. This is in line with the observations of no pronounced twist of electron-microscopy-reconstructed HeLa spindles and less or no twist in non-cancer RPE1 cells (Kiewisz et al, 2022; Trupinić et al, 2022; Neahring et al, 2021). Hence, rather than exclusively generating spindle twist, KIF11 might regulate and balance the torques, allowing for flexible, context-dependent filament organization.

## Methods

### Cloning, gene expression, and protein purification

The gene of KIF11 was cut with NotI and AscI and inserted into an OCC vector with or without C-terminal EGFP and with a $His_6$ tag, separated by a 3C protease cleavage site. For mutagenesis, the two or four C-terminal residues of the neck linker were removed (NL16 and NL14) or a GS (NL20) or GSGS (NL22) sequence was added at the C-terminus. The above vector was PCR amplified with respective primers: for the shortened constructs the forward primer started after the 3′ end of the neck linker and the reverse primer started at the 5′ end at amino acid residue 14 or 16 of the neck linker. For the elongated constructs, the forward primer started after the 3′ end of the neck linker and bore the respective insert (Appendix Tab. S2). The reverse primer started at the 5′ end of the neck linker. For cloning, *Escherichia coli* DH5α was used.

Viruses were generated using the FlexiBac system (Lemaitre et al, 2019). SF9 cells (IPLB Sf21-AE, Merck 71104) at 1 million cells per mL were infected with virus (1:100, v/v) and genes were expressed for 96 h at 27 °C and 120 rpm. Cells were centrifuged with $300 \times g$ for 10 min at 4 °C. Pellets were resuspended in PBS (1% of expression volume) with protease inhibitor, flash frozen in liquid nitrogen and stored at −80 °C. For purification, cell pellets were thawed on ice and resuspended in purification buffer (50 mM $NaH_2PO_4$, 300 mM KCl, 2 mM $MgCl_2$, 1 mM DTT, 0.1 mM ATP, pH 7.5) with protease inhibitor. The lysate was cleared with an ultracentrifuge spin with 40,000 rpm for 1 h at 4 °C. The supernatant was filtered through a 0.45 μm filter and loaded on a 1 mL HiTrap column with a superloop. The column was washed with immobilized metal affinity chromatography (IMAC) wash buffer (purification buffer with 20 mM imidazole) and the protein was eluted with IMAC elution buffer (purification buffer with 300 mM imidazole) with an elution gradient. Protein-containing fractions were pooled and concentrated with Amicon filters (cutoff 100 kDa). 3C protease was added (1:150, v/v) and the $His_6$ tag was cleaved over night at 4 °C. The protein solution was diluted 6-fold to reduce the imidazole concentration and passed over the HiTrap column again. The protease remained bound to the column with its $His_6$ tag. The flow through was concentrated to 0.5 mL, cleared at $17,000 \times g$ for 10 min, and gel filtered over a Superose6 column with purification buffer. 5% glycerol was added and the protein was flash-frozen in liquid nitrogen and stored at −80 °C (Appendix Fig. S1).

## Microtubule polymerization

Tubulin was purified from pig brains according to standard protocols (Castoldi and Popov, 2003) and labeled in house. Briefly, tubulin was polymerized and microtubules were isolated by centrifugation in a glycerol cushion. Microtubules were mixed with the labeling dye (TAMRA succinimidyl ester, ThermoFischer; Alexa Fluor 488 Carboxylic Acid, 2,3,5,6-Tetrafluorophenyl ester, ThermoFischer; ATTO 647N Amine-reactive NHS-ester, ATTO-TEC) in a 10 to 20-fold molar excess of the dye and incubated for 30 min. The reaction was quenched with 0.5 M potassium glutamate and the tubulin was purified with depolymerization-polymerization cycles. Labeled tubulin was mixed in a ratio of 1:3 with unlabeled tubulin for experiments. Fixed and cargo microtubules were grown with tubulin with guanylyl-($\alpha,\beta$)-methylene-diphosphonate (GMP-CPP) and stabilized with taxol. For *fixed microtubules*, 40 µL elongation mix containing 1.25 mM GMP-CPP, 1.25 mM $MgCl_2$ and 4.5 µM TAMRA labeled tubulin in BRB80 (80 mM Pipes at pH 6.9, 1 mM $MgCl_2$, 1 mM EGTA) was incubated on ice for 5 min and then for 30 min at 37 °C. Microtubules were pelleted (17,000 × $g$, 15 min) and resuspended in elongation mix (1.25 mM GMP-CPP, 1.25 mM $MgCl_2$ and 0.5 µM TAMRA labeled tubulin in BRB80) and grown for 2–3 days at 32 °C. Microtubules were pelleted (17,000 × $g$, 8 min) and gently resuspended in BRB80 with 10 µM taxol (BRB80X). They were kept at room temperature for several days for annealing. To grow *cargo microtubules*, a polymerization mix with 1.25 mM GMP-CPP, 1.25 mM $MgCl_2$ and 5 µM Atto647N labeled tubulin was in incubated on ice for 5 min and then for 8 min at 37 °C. Microtubules were pelleted (17,000 × $g$, 15 min) and resuspended in BRB80X.

## Fabrication and treatment of ridge structures

Ridge structures were produced and coated with dichlorodimethyl-silane (DDS) as described in Mitra et al (Mitra et al, 2018; Mitra et al, 2020).

## KIF11-driven microtubule sliding assays

To assemble 6 flow chambers, 7 strips of Nescofilm were placed on the coverslip with structures and covered with a regular coverslip with DDS coating. The Nescofilm was melted on a hot plate. Channels were flushed with: (i) 1:60 TetraSpeck bead solution in PBS (v/v, 200 nm) for 2 min, (ii) PBS wash, (iii) 0.2 mg/mL TAMRA 5G5 antibody (Invitrogen, RRID AB_2536728) solution in PBS for 5 min, (iv) 1% F127 (w/v in PBS) solution for at least 1 h, (v) 3 washes with BRB80, (vi) fixed microtubules in motility buffer (MB-ADP, BRB80 with 10 µM taxol, 200 µg/mL casein (from bovine milk, Sigma C7078), 10 mM DTT, 0.1% (v/v) Tween-20, 20 mM D-glucose, 1 mM ADP), (vii) MB-ADP wash, (viii) 10 nM KIF11-EGFP in MB-ADP for 5 min, (ix) MB-ADP wash, (x) cargo microtubules in MB-ADP, (xi) wash with MB-ADP + + (MB-ADP with 200 µg/mL glucose oxidase and 20 µg mL$^{-1}$ catalase), (xii) MB-ATP + + (MB-ADP + + with 1 mM ATP instead of ADP). Where indicated, the final ATP concentration was reduced to 250, 200, 150, 30, and 25 µM. In the buffer conditions test, BRB40 (40 mM Pipes at pH 6.9, 0.5 mM $MgCl_2$, 0.5 mM EGTA) or Hepes (20 mM Hepes at pH 7.2 with 50 mM KCl, 2 mM $MgCl_2$, 1 mM EGTA) based motility buffers were used.

To determine the helicity of cargo microtubule motion, we performed control experiments. We lowered the focal plane so that TetraSpeck beads in valleys were in focus—thus, the focal plane was about 360 nm below the fixed microtubules. This way, the fluorescence signal intensity of the cargo microtubules increased monotonically when the cargo microtubule went from the top of the fixed microtubule to the bottom. The handedness was then obtained by comparing the signal intensity to the sideways distance and is given as viewed from the minus end of the fixed microtubule. In case of a right-handed helical motion or orbiting, a maximum of the signal intensity was followed by a maximum of the sideways distance (with a shift of λ/4). This corresponds to a motion from the bottom to the left of the fixed microtubule.

Data was acquired on 1–2 independent experimental days, in at least 2 different channels (except 25 and 250 µM ATP and Hepes-based motility buffer).

## Polarity-labeling of microtubules

For polarity labeling, the plus end of Atto647N labeled cargo microtubules was elongated with Atto488 labeled tubulin, which was partially labeled with N-Ethylmaleimide (NEM) to suppress minus end growth. Based on Phelps et al, a maleimidation mix (10 µM Atto488 tubulin, 0.4 mM NEM and 1 mM GTP in BRB80) was incubated for 10 min on ice and the reaction was quenched with 20 mM DTT for at least 10 min on ice (Phelps and Walker, 2000). An elongation mix (4 mM $MgCl_2$, 1 mM GTP, 1 µM Atto488 NEM tubulin, and 4 µM Atto488 tubulin in BRB80) was incubated for 5 min on ice. The elongation mix was preheated for 30 s at 37 °C and cargo microtubules were added (tubulin concentration 1 µM). Microtubules were elongated for 5 min at 37 °C, incubated with 5-fold excess BRB80 with 20 µM taxol for 1 min at room temperature, centrifuged at 17,000 × $g$ for 15 min, and resuspended in BRB80X. This resulted in Atto647N labeled cargo microtubules with a short (<2 µm) Atto488 labeled plus end. Polarity-labeled microtubules with Atto488 extensions were also used as fixed microtubules to confirm the pronounced end residency and accumulation of KIF11-EGFP at plus ends. To determine the plus end of the fixed microtubule, microtubules were imaged 1 h after the initiation of sliding in the GFP channel.

## Optical image acquisition

Optical imaging was performed using an inverted fluorescence microscope (Axio Observer Z1; Carl Zeiss Microscopy GmbH) with a 63× oil immersion 1.46NA objective (Zeiss) in combination with an EMCDD camera (iXon Ultra; Andor Technology) controlled by Metamorph (Molecular Devices Corporation). A LED white light lamp (Sola Light Engine; Lumencor) in combination with a TRITC filterset (ex 520/35, em 585/40, dc 532: all Chroma Technology Corp.), an Atto647N filterset (ex 628/40, em 692/40, dc 635) and a GFP filter set (ex 475/35, em 525/45), corresponding to TAMRA labeled microtubules, Atto647N labeled microtubules and EGFP labeled motors/Atto488 labeled microtubule extensions, respectively, were used for epifluorescence imaging. The imaging temperature was maintained at 24 °C by fitting a custom-made hollow brass ring around the body of the objective and connecting it to a water bath with a cooling/heating unit (F-25-MC Refrigerated/Heating Circulator; JULABO GmbH). The sliding of

cargo microtubules was imaged in the Atto647N channel for 10 min with 10 frames/s with an exposure time of 100 ms. Fixed microtubules were imaged in the TRITC channel for 200 frames at 10 frames/s with an exposure time of 100 ms after imaging the cargo microtubules.

## Image processing and data analysis

Data was processed and analyzed as described in Mitra et al (Mitra et al, 2020). Positions of fixed and cargo microtubules as well as TetraSpeck beads (in TAMRA and Atto647N channel) were obtained from the MATLAB-based tracking software FIESTA (Ruhnow et al, 2011). The TetraSpeck beads were used to correct the microtubule tracks for drift and color offset. For the KIF11-EGFP NL18 construct, the pitches were typically smaller than 2.5 μm and only tracks over valley regions were analyzed. For some of the neck linker mutants, a number of the pitches were larger than 5 μm. In those tracks, the ridges (width of 2–5 μm) were often not clearly visible in the measured trajectories, probably because the cargo microtubules could undergo up to half a rotation on an individual ridge. Thus, in these cases we did not distinguish between ridge and valley regions. The fixed microtubules were imaged over 200 frames and the tracked positions were averaged to obtain the filament position. The distance of the cargo microtubule's center point to the fixed microtubule averaged center line was calculated. Negative sideways distances were assigned to cargo microtubules moving in the obtained images on the right side of the fixed microtubule (as viewed from the trailing end of the cargo microtubule which is identical to viewing from the minus end of the fixed microtubule). When no averaged positions of the fixed microtubules were obtainable (e.g., because of interfering signals from small microtubule aggregates close to the fixed microtubule, two fixed microtubules too close together to be resolved or movement of the fixed microtubule between imaging of the cargo and fixed microtubule), the path of the cargo microtubule was averaged over 10 μm and the sideways distance was calculated with respect to the averaged path. For display, cargo microtubule tracks were smoothed over 50 frames.

To obtain the motility parameters with manual computer-aided measurements, the sideways distance was plotted over time and minima and maxima of the rotations were marked. From these points the motility parameters (forward velocity = forward distance of the cargo microtubule along the direction of the fixed microtubule per time, angular velocity = $2\pi$ divided by the time per rotation, and pitch = forward distance traveled per full rotation) were calculated for each rotation and averaged for each cargo microtubule. For ridge-valley comparisons, the forward velocity was calculated the following way: the first and last 50 frames of the time and forward distance of each cargo microtubule track were averaged separately. The difference in distance divided by the difference in time yielded the forward velocity. The fluorescence intensities of the cargo microtubules were obtained from the tracking data of the software FIESTA (Ruhnow et al, 2011).

We considered cargo microtubules as sideways-only when their forward velocity was 20% or less than the mean velocity of forward-and-sideways cargo microtubules. This cutoff was set based on the bimodal, clearly separated velocity distribution (Fig. 4B, black and purple). Further we note, that forward-only microtubules and stuck microtubules could actually be forward-

and-sideways microtubules and sideways-only microtubules, respectively, for which the sideways motion was not detected. A technical reason for not detecting the sideways motion could have been an unprecise cargo microtubule tracking due to too jerky or erratic motion. We observed microtubules of all four categories on the same fixed microtubule and thus, can rule out that the fixed microtubule determines the category of the cargo microtubule.

## Statistical analysis

Motility parameters were calculated as mean ± standard deviation. For ridge-valley transitions, a Student's t-Test was used to compare the data to 1 (ridge to valley: $p = 0.28$, valley to ridge: $p = 0.18$).

## Data availability

This study includes no data deposited in external repositories.

## Peer review information

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

## Acknowledgements

We thank Corina Bräuer for her technical support as well as Ludger Santen, Iva Tolić, Nenad Pavin, Sophie Dumont and all members of the Diez laboratory for scientific discussions. We acknowledge Régis Lemaitre and the Protein Expression, Purification, and Chromatography Facility of MPI-CBG Dresden for generating the baculovirus. We thank the Microstructure Core Facility of the Technology Platform of the Center for Molecular and Cellular Bioengineering at TU Dresden for providing the ridge micro-structures. The Core Facility is supported by the Deutsche Forschungsgemeinschaft (DFG, German Research Foundation) under Germany's Excellence Strategy – EXC-2068 – 390729961 – Cluster of Excellence Physics of Life of TU Dresden. We would like to acknowledge funding from the German Research Foundation (SFB1027), the Boehringer Ingelheim Fonds (PhD stipend to LM), and the Free State of Saxony (PhD stipend to LN).

## Author contributions

**Laura Meißner**: Conceptualization; Resources; Data curation; Formal analysis; Supervision; Funding acquisition; Validation; Investigation; Visualization; Methodology; Writing—original draft. **Lukas Niese**: Data curation; Formal analysis; Investigation; Writing—review and editing. **Irene Schüring**: Data curation; Formal analysis; Writing—review and editing. **Aniruddha Mitra**: Conceptualization; Writing—review and editing. **Stefan Diez**: Conceptualization; Resources; Supervision; Funding acquisition; Investigation; Methodology; Project administration; Writing—review and editing.

## Funding

## Disclosure and competing interests statement

The authors declare no competing interests.

