## [Peer Review File · The EMBO Journal]

Kinesin-5 drives the helical motion of anti-parallel and parallel microtubules around each other

Laura Meißner, Lukas Niese, Irene Schüring, Aniruddha Mitra, and Stefan Diez

Corresponding author(s): Stefan Diez (stefan.diez@tu-dresden.de)

Review Timeline:

Submission Date:	27th Jul 23
Editorial Decision:	1st Sep 23
Revision Received:	30th Dec 23
Editorial Decision:	22nd Jan 24
Revision Received:	24th Jan 24
Accepted:	29th Jan 24

Editor: Hartmut Vodermaier

Transaction Report:

Dr. Stefan Diez
TU Dresden
B CUBE
Tatzberg 41
Saxony 01307
Germany

1st Sep 2023

Re: EMBOJ-2023-115116
Kinesin-5 drives the helical motion of anti-parallel and parallel microtubules around each other

Dear Stefan,

Thank you again for submitting your study on kinesin-5 lateral motility and microtubule rotation to The EMBO Journal. I have now received the reviews of four expert referees, copied below for your information, and I am happy to say that all reviewers found this work interesting and generally well-conducted. They nevertheless raised a couple of specific issues that would need to be adequately addressed prior to publication.

I am therefore inviting you to prepare a revised version along the lines suggested in the four reports. Since it is our policy to consider only a single round of major revision and therefore important to fully answer to all comments at the time of resubmission, please do not hesitate to get back to me with a tentative response letter/revision plan, in case you would like to clarify/discuss certain points already during the early stages of the revision. I should add that we could also offer extension of the default three-months revision period if needed, with our 'scooping protection' (meaning that competing work appearing elsewhere in the meantime will not affect our considerations of your study) remaining of course valid also throughout this extension.

Detailed information on preparing, formatting and uploading a revised manuscript can be found below and in our Guide to Authors. Thank you again for the opportunity to consider this work for The EMBO Journal, and I look forward to your revision in due time.

With best regards,

Hartmut

9) Digital image enhancement is acceptable practice, as long as it accurately represents the original data and conforms to community standards. If a figure has been subjected to significant electronic manipulation, this must be clearly noted in the figure legend and/or the 'Materials and Methods' section. The editors reserve the right to request original versions of figures and the original images that were used to assemble the figure. Finally, we generally encourage uploading of numerical as well as gel/blot image source data; for details see: embopress.org/page/journal/14602075/authorguide#sourcedata

At EMBO Press, we ask authors to provide source data for the main manuscript figures. Our source data coordinator will contact you to discuss which figure panels we would need source data for and will also provide you with helpful tips on how to upload and organize the files.

In the interest of ensuring the conceptual advance provided by the work, we recommend submitting a revision within 3 months (30th Nov 2023). Please discuss the revision progress ahead of this time with the editor if you require more time to complete the revisions. Use the link below to submit your revision:

Link Not Available

Referee #1:

The authors show here that side-stepping of kinesin-5 KIF11 causes sliding microtubule pairs that are connected by this motor to rotate around each other. Experiments are performed using the in vitro assay previously developed by the Diez lab to show side stepping of processive motors (e.g. Mitra 2018) and similar microtubule rotations driven by kinesin14 Ncd (Mitra 2020). It is a technically very elegant assay that allows to determine the handedness of the motion of a cargo microtubule around an immobilized, suspended microtubule and the distance between these two microtubules with high precision. Multiple studies have shown side-stepping of motors previously, this study is now the second example demonstrating helical motor-driven movements for antiparallel microtubules. A nice result that goes beyond the previous work is the observation that the distance between microtubules connected by KIF11 depends on their orientation: for parallel microtubules not sliding relative to each other (but still rotating), the distance corresponds to the length of KIF11, but this distance is shorter for antiparallel, sliding microtubules, suggesting that the motors do now 'tilt' as the transport microtubule experiences viscous drag. These numbers could be quite useful for computational studies of spindle structure and/or motor competition, and could maybe also be compared to microtubule distances in recent EM studies of spindles. Overall the experiments are very carefully performed and clearly presented and further characterize the mechanical activity of this important spindle motor. Side stepping may be of interest for understanding the structure of the spindle, given 'twisted' microtubules were recently observed in spindles, as mentioned also in the Discussion of this manuscript.

Specific comments:

1. The KIF11 construct used here is tagged with GFP at the C-terminus, but GFP fluorescence is not used in the experiments.

Given that recent work from the Al-Bassam lab (Bodrug 2020) showed that the C-terminal tail of KIF11 affects the biochemical properties of the motor domains, it would be comforting to know that helical sliding motions of microtubules can also be observed with untagged KIF11.

2. To which extent is the helical motion dependent on the buffer composition? The Bodrum paper showed that changing the salt concentration can strongly affect KIF11 speed and might also affect side stepping. Testing the effect of the ionic strength on side stepping would add novelty to the study.

3. This reviewer has difficulties to recognize the polarity marking of the polarity-marked microtubules in Fig. S3. Some more convincing images should be shown. It would also be useful to report how the quality of the polarity marking was evaluated. This seems important, because the distance analysis in Fig. 4 and the interpretation that rotating, non-sliding microtubules are parallel -both major novelties of the study - depend on good quality polarity marks.

Minor comments:

4. The Methods sometimes lack detail, examples: which "kit" was used for tubulin labelling, which type of casein was used, in which buffers were KIF11 and cargo microtubules added to the assay?

5. Why was detergent present in the final assay buffer, this appears unusual - could it affect the motile behavior of KIF11?

6. The Discussion is pleasantly comprehensive, but can probably be streamlined.

Referee #2:

In this elegant in vitro study, the authors investigated microtubule transport driven by the motor protein kinesin-5. They investigated the movement of microtubules in 3 dimensions, which allowed them to quantify the forward and lateral components of microtubule movement. For this purpose, they used a three-dimensional motility assay, which was recently developed by the same laboratory. The authors investigated how free microtubules move with respect to those fixed on the surface, when they are oriented anti-parallel and parallel, and for several different motor constructs. In the case of anti-parallel microtubule configuration, the motor generated microtubule sliding and right-handed rotation around each other. Surprisingly, rotational motion was also observed in the case of parallel microtubules. Finally, the presentation of these results is clear, the manuscript is carefully written and a pleasure to read.

I have two suggestions for improving the manuscript, which can be addressed textually.

(1) The authors analyze only microtubules with at least two full rotations. However, this choice could affect the pitch value. For example, in their 10 μm wide valleys, a pitch value greater than 5 μm cannot be determined, because in this case the microtubules will not have enough space to complete two full rotations. It is additionally confusing that some measured pitch values exceed 5 μm . Please clarify this inconsistency.

(2) The rotational movement of microtubules in the case of a parallel configuration is somewhat surprising and at first glance it is not clear how the symmetry breaking occurs. In the experiments here, the experimental setup in which one microtubule is fixed while the other is free to move along the fixed one breaks the symmetry. However, it is unclear how sidestepping could affect a system of several parallel microtubules with fixed minus ends. It would be interesting to speculate on such possibility because it corresponds to living systems in which microtubules are attached by their minus ends to microtubule-organizing centers.

Referee #3:

In this manuscript, Meisner and colleagues present an intriguing study on the lateral motility of human kinesin-5. The ability of microtubule-dependent motors to execute lateral movements has been recognized for some time, with various laboratories and assays demonstrating this phenomenon. However, the underlying significance of this behavior remains elusive. While the current manuscript does not provide a definitive explanation for the mechanics behind kinesin-5's lateral movements, it does offer a fresh perspective on the intricacies of this phenomenon.

This study builds upon the authors' previous publication, wherein a similar microscopy-based assay employing suspended microtubules was employed to investigate the stepping behavior of kinesin 14. The experimental approach is novel, executed with precision, and the results are well documented. These findings are poised to raise interest among researchers specialized in kinesin biology.

Nonetheless, a noteworthy caveat necessitates attention before the paper can be deemed ready for publication based on its current conclusions. Within the present manuscript, the researchers introduce perturbations to the kinesin-microtubule motility

system using diverse techniques: manipulation of ATP concentration, mutations in the neck linker, and manipulation of microtubule polarity represent some of the key experimental parameters examined. Consequently, alterations in motility metrics such as rate, pitch, and stepping probability are ascribed to these interventions. However, a relatively unexplored factor pertains to changes in motor density among overlapping microtubules.

As indicated by the authors (on page 8), disparate motor densities along the microtubule walls and tips could potentially contribute to the pronounced variability in observable behaviors. This leads to the concern that interventions such as varying ATP concentration or introducing neck linker mutations might indirectly influence lateral movements by modulating the number of engaged motors. Addressing this concern directly entails two approaches: firstly, varying motor concentration in the original assays utilizing wild-type motors and 1 mM ATP; secondly, visualizing motor density through employment of GFP-tagged motors on overlapping microtubules or at least on immobilized unbundled microtubules on coverslips.

The outcome of these supplementary assays will serve to either dispel the "affinity" concern by demonstrating that motor density in isolation cannot account for the observed motility changes, or potentially prompt an adjusted discussion that duly acknowledges this multifaceted complexity.

For example:

Current authors conclusion for ATP experiments: " This indicates, that the slower a cargo microtubule moves in the longitudinal direction, the more likely it moves sideways in the axial direction."

Possible explanation 2. If fewer motors are present at lower ATP, could this explain slower forward and sideways motility? The fact that the side stepping probability is increased at lower speeds is not disputed but the cause may be different than suggested

Possible explanation 3: if more motor molecules accumulate at the microtubule tip during faster motility (or for different mutants), could this impede rotational stepping, thereby lowering its apparent probability?

Minor comments

1. Please describe the KIF11 construct used in this work early in the results section.

2. The units for stepping probability are not intuitive. Please consider using fractions of 1 and clearly describe what this probability reflects

3. Discussion

"Perturbing the neck linker length altered the motility parameters, indicating the KIF11 has evolved for slow and robust motility"

Please modify. This study does not address evolution of KIF11 or robustness of its motility. Perhaps the case can be made that the linker is optimized for certain features, but the statement would benefit from evidence-based specificity

(Also in abstract) "We conclude that the helical motion of microtubules driven by KIF11 allows for flexible, context-dependent filament organization and torque regulation in the mitotic spindle." The fact that the helicity has different polarity in cells and in vitro is very intriguing but it hardly allows to conclude that KIF11 regulates the torque or it is involved in "flexible, context-dependent filament organization"

4. Introduction:

Page 3. "Recently, high resolution imaging of HeLa and RPE1 cells revealed, that the spindle is twisted into a chiral structure."

Please improve writing, this statement contradicts literature cited in this paper, including the sentence in the same paragraph: "In contrast to HeLa cells, spindles of RPE1 cells did not exhibit helicity ..."

Page 3. "This suggests, that KIF11 has evolved as a slowly sliding, fast rotating motor ..". Please modify. This work has no bearing on KIF11 motor evolution. Also, the rates of sliding and rotation are provided in this work in different units (page 4).

Please supplement this or similar sentence with numerical results using the same units (perhaps steps per sec)

5. Figure 3. Is directional velocity zero for all microtubules? If not, would it be possible to provide actual data? Figure 6 B,C is difficult to follow and it does not add much to discussion because this aspect seems very speculative and other features of the system may be at play. A cartoon or table that summarizes differences/similarities in sidestepping of different motors would enhance this last figure, but this is just a suggestion for authors.

Referee #4:

Recently, kinesin-5 has been implicated in the generation of microtubule twisting at the mitotic spindle microtubules. Meißner et al. study the helical motion of the cargo microtubule driven by tetrameric human kinesin-5 KIF11, which is an important player in prometaphase, along the freely suspended, fixed microtubule in vitro. By applying the experimental setups from their previous work on the kinesin-14 motor proteins, the authors found that tetrameric plus-end-directed kinesin-5, KIF11, slides the cargo microtubules around the fixed microtubule in a right-handed helical motion. The pitch of the helical motion depends on the ATP concentration, indicating changing the ratio of forward step to side step. Furthermore, this is a novel finding that extension of the KIF11 controls the forward velocity of antiparallel cargo microtubules. Overall, this study led to important new knowledge about the motility and regulation of torque component of KIF11 in MT-MT sliding. I support the publication of this work with several revisions.

My comments are as follows:

1)The authors state in the abstract: "~and found that the motor caused right-handed rotation of anti-parallel microtubules around each other." However, they did not observe the rotational motion of the cargo MT itself along its longitudinal axis. It is better to delete the description of the direction of rotation of the cargo microtubule along its long axis. In addition, in Fig 6A, authors show the direction of the microtubule rotation in the microtubule cross-section as a purple arrow, and the direction of rotation of kinesin as it moves around the microtubule surface as a gray arrow. These arrows need to be removed from the figure or the results need to be provided for the rotational direction of the sliding cargo MT itself along its long axis, driven by cross-linking KIF11 molecules.

2)My main concern is that in Figure 3D, the helix direction appears to be left-handed, contrary to Figure 1C. Even considering the direction of the axis indicating light intensity and sideways distance, it appears to be left-handed. Although the parallel cargo microtubules are stated to orbit without forward motion, the cargo microtubule shown in the associated movies (S2 and S4) clearly appears to move forward along a fixed microtubule, i.e., helical motion. Movie 4 also shows that the cargo microtubule clearly moves forward along a fixed microtubule at several nm/s toward the minus end of the microtubule. Perhaps the plot shown in the figure 3D is for a parallel cargo microtubule moving (slowly) toward the minus end of a fixed microtubule, in which case it seems to have the same helical motion as a minus-end-directed kinesin-14 in a right-handed helical motion. If the authors intend to claim right-handed orbiting (counterclockwise rotation without forward motion) as seen from the minus end of the microtubule, a plot and its movie of the case with almost no forward motion, equivalent to Figure 3D, is needed. Also, it would be better to show the plot and its movie of the slow movement of a parallel cargo microtubule toward the plus end of a fixed microtubule as supplementary information.

3)It is better to show which end of the fixed microtubule is the + or - end in Movie 2

4)There are several issues regarding the description of the number of samples in the experiments. For instance, in Figure 2, the number of ATP concentration-dependent assays is not provided. Additionally, there is a discrepancy in the number of "sideways-only polarity-labeled cargo microtubules", which are stated as 18 and 9 in the same section. Can you clarify the difference between these two numbers? Also, while the representation of "motor forward" microtubules are referred to as both 47% and 306 events, for a total of 651 events, the representation of "immobile in forward" microtubules are presented as 53% and 341 events, for a total of 643 events. Can you explain the reason for the difference in numbers? For clarity, please provide a clear breakdown of the number of samples in the experiments. This will help to understand the details.

We thank all reviewers for their careful and constructive review of our manuscript. Please find below our detailed point-to-point responses to the reviewers' comments and our actions taken. In addition, the key changes are also indicated by blue color in a submitted version of the manuscript for review only.

In particular, we (i) performed additional experiments and analyses: we repeated experiments with unlabeled KIF11 (new Appendix Fig. 2) and tested the effect of motor concentration as well as buffer composition on motility parameters of KIF11 (new Appendix Figs. 4 and 5), (ii) added clarifications (e.g. modified Appendix Fig. 6, Main Text and Materials and Methods) and (iii) extended the discussion of our results in the Main Text.

Referee #1:

The authors show here that side-stepping of kinesin-5 KIF11 causes sliding microtubule pairs that are connected by this motor to rotate around each other. Experiments are performed using the in vitro assay previously developed by the Diez lab to show side stepping of processive motors (e.g. Mitra 2018) and similar microtubule rotations driven by kinesin14 Ncd (Mitra 2020). It is a technically very elegant assay that allows to determine the handedness of the motion of a cargo microtubule around an immobilized, suspended microtubule and the distance between these two microtubules with high precision. Multiple studies have shown side-stepping of motors previously, this study is now the second example demonstrating helical motor-driven movements for antiparallel microtubules. A nice result that goes beyond the previous work is the observation that the distance between microtubules connected by KIF11 depends on their orientation: for parallel microtubules not sliding relative to each other (but still rotating), the distance corresponds to the length of KIF11, but this distance is shorter for antiparallel, sliding microtubules, suggesting that the motors do now 'tilt' as the transport microtubule experiences viscous drag. These numbers could be quite useful for computational studies of spindle structure and/or motor competition, and could maybe also be compared to microtubule distances in recent EM studies of spindles. Overall, the experiments are very carefully performed and clearly presented and further characterize the mechanical activity of this important spindle motor. Side stepping may be of interest for understanding the structure of the spindle, given 'twisted' microtubules were recently observed in spindles, as mentioned also in the Discussion of this manuscript.

Response: We thank the reviewer for the positive assessment and appreciation of our work as well for the valuable comments. Please see for our specific responses below.

Specific comments:

1. The KIF11 construct used here is tagged with GFP at the C-terminus, but GFP fluorescence is not used in the experiments. Given that recent work from the Al-Bassam lab (Bodrug 2020) showed that the C-terminal tail of KIF11 affects the biochemical properties of the motor domains, it would be comforting to know that helical sliding motions of microtubules can also be observed with untagged KIF11.

Response: We performed our experiments with an EGFP (C-terminal) KIF11 labeled construct because we intended to use the GFP fluorescence as a measure of motor density. However,

the autofluorescence of the ridges did not allow conclusions about the motor density. As control, we now performed additional 3D sliding assays using KIF11 without a fluorescent tag. **We added this data as Appendix Fig. 2 and discuss it in the Main Text as well as in the Materials and Methods.** The unlabeled construct also drove a robust helical motion of cargo microtubules around fixed microtubules, though with slightly different motility parameters (reduced forward velocity but similar pitch) compared to the EGFP construct.

2. To which extent is the helical motion dependent on the buffer composition? The Bodrug paper showed that changing the salt concentration can strongly affect KIF11 speed and might also affect side stepping. Testing the effect of the ionic strength on side stepping would add novelty to the study.

Response: We performed new experiments with different buffers: we halved the ionic strength (BRB40) and used a Hepes based buffer (as in the Bodrug paper) with similar ionic strength as BRB80. In Hepes based buffer, motility parameters of cargo microtubules driven by KIF11 slightly increased. **We added these results in Appendix Fig. 5 and described them in the Main Text.**

3. This reviewer has difficulties to recognize the polarity marking of the polarity-marked microtubules in Fig. S3. Some more convincing images should be shown. It would also be useful to report how the quality of the polarity marking was evaluated. This seems important, because the distance analysis in Fig. 4 and the interpretation that rotating, non-sliding microtubules are parallel -both major novelties of the study - depend on good quality polarity marks.

Response: We determined the polarity and orientation of the microtubules the following way: the cargo microtubules were labeled with Atto488 extensions. The use of N-Ethylmaleimide suppressed minus end extension, resulting in cargo microtubules with a labeled plus end of up to a few μm (usually $< 2 \mu\text{m}$). We only used such dual-color cargo microtubules for our analysis. To determine the polarity of the fixed microtubules, we used the plus-end residency of KIF11-EGFP. The fixed microtubules were imaged one hour after starting the sliding process, and the plus end was visible by a blob of EGFP signal. We only used events, where this end accumulation was clearly visible. In total we evaluated 23 events, where the labels of both microtubules were clearly visible (as described above) so we are 100% sure of their polarity (yielding five anti-parallel and 18 parallel overlaps). **We clarified this in the Materials and Methods. We modified Appendix Fig. 6 (former Fig. S3) by adding separate images for each fluorescence channel.**

Minor comments:

4. The Methods sometimes lack detail, examples: which "kit" was used for tubulin labelling, which type of casein was used, in which buffers were KIF11 and cargo microtubules added to the assay?

Response: We used casein from bovine milk (Sigma C7078). The default assay buffer for the solutions was MB-ADP (BRB80 with 10 μM taxol, 200 $\mu\text{g/mL}$ casein, 10 mM DTT, 0.1% (v/v) Tween-20, 20 mM D-glucose, 1 mM ADP). **We added a more detailed description of the used buffers and the tubulin labeling procedure in the SI.**

5. Why was detergent present in the final assay buffer, this appears unusual - could it affect the motile behavior of KIF11?

We used detergent in the assay buffer to avoid unspecific binding of cargo microtubules to the surface – as before in Mitra et al. 2020. Additionally, detergent counteracts the tendency of KIF11 to form clusters. We now performed additional control experiments with a BRB80 based motility buffer without Tween-20. Cargo microtubules driven by KIF11 showed similar motility parameters as in presence of Tween-20. **We added these results in Appendix Fig. 5 and described them in the Main Text.**

6. The Discussion is pleasantly comprehensive, but can probably be streamlined.

Response: We followed the suggestion by the reviewer and streamlined the Discussion without compromising on content.

Referee #2:

In this elegant in vitro study, the authors investigated microtubule transport driven by the motor protein kinesin-5. They investigated the movement of microtubules in 3 dimensions, which allowed them to quantify the forward and lateral components of microtubule movement. For this purpose, they used a three-dimensional motility assay, which was recently developed by the same laboratory. The authors investigated how free microtubules move with respect to those fixed on the surface, when they are oriented anti-parallel and parallel, and for several different motor constructs. In the case of anti-parallel microtubule configuration, the motor generated microtubule sliding and right-handed rotation around each other. Surprisingly, rotational motion was also observed in the case of parallel microtubules. Finally, the presentation of these results is clear, the manuscript is carefully written and a pleasure to read.

Response: We thank the reviewer for the positive assessment and appreciation of our work as well for the valuable comments. Please see for our specific responses below.

I have two suggestions for improving the manuscript, which can be addressed textually.

(1) The authors analyze only microtubules with at least two full rotations. However, this choice could affect the pitch value. For example, in their 10 μm wide valleys, a pitch value greater than 5 μm cannot be determined, because in this case the microtubules will not have enough space to complete two full rotations. It is additionally confusing that some measured pitch values exceed 5 μm . Please clarify this inconsistency.

Response: Well spotted. Pitches are typically smaller than 2.5 μm for the NL18 KIF11-EGFP construct. However, we indeed observed pitches larger than 5 μm for some of the neck linker mutants. In tracks with such large pitches, the ridges (width of 2-5 μm) were not clearly visible in the trajectories, probably because cargo microtubules can undergo up to half a rotation on an individual ridge. Thus, in these cases we did not distinguish between ridge and valley regions. **We now clarify this in the Materials and Methods.**

(2) The rotational movement of microtubules in the case of a parallel configuration is somewhat surprising and at first glance it is not clear how the symmetry breaking occurs. In the experiments here, the experimental setup in which one microtubule is fixed while the other is free to move along the fixed one breaks the symmetry. However, it is unclear how sidestepping could affect a system of several parallel microtubules with fixed minus ends. It would be interesting to

speculate on such possibility because it corresponds to living systems in which microtubules are attached by their minus ends to microtubule-organizing centers.

Response: The reviewer raises an interesting point. In our setup, the fixed microtubule is fixed on both ends, whereas both ends of the cargo microtubule are free. In a microtubule overlap in the spindle, one end of each microtubule is fixed – which is a different geometry. In this scenario we hypothesize that anti-parallel microtubules roll against each other while sliding apart (leading to a supertwist of the pair) and parallel microtubules twist themselves (without generating a supertwist of the pair). However, this most likely strongly depends on their anchorage, e.g., how rigidly microtubules are bound at the MTOC, and how passive cross-linkers stabilize the microtubule bundles. **We added this hypothesis to the Discussion.**

Referee #3:

In this manuscript, Meisner and colleagues present an intriguing study on the lateral motility of human kinesin-5. The ability of microtubule-dependent motors to execute lateral movements has been recognized for some time, with various laboratories and assays demonstrating this phenomenon. However, the underlying significance of this behavior remains elusive. While the current manuscript does not provide a definitive explanation for the mechanics behind kinesin-5's lateral movements, it does offer a fresh perspective on the intricacies of this phenomenon. This study builds upon the authors' previous publication, wherein a similar microscopy-based assay employing suspended microtubules was employed to investigate the stepping behavior of kinesin 14. The experimental approach is novel, executed with precision, and the results are well documented. These findings are poised to raise interest among researchers specialized in kinesin biology.

Response: We thank the reviewer for the positive assessment and appreciation of our work as well for the valuable comments. Please see for our specific responses below.

Nonetheless, a noteworthy caveat necessitates attention before the paper can be deemed ready for publication based on its current conclusions. Within the present manuscript, the researchers introduce perturbations to the kinesin-microtubule motility system using diverse techniques: manipulation of ATP concentration, mutations in the neck linker, and manipulation of microtubule polarity represent some of the key experimental parameters examined. Consequently, alterations in motility metrics such as rate, pitch, and stepping probability are ascribed to these interventions. However, a relatively unexplored factor pertains to changes in motor density among overlapping microtubules.

As indicated by the authors (on page 8), disparate motor densities along the microtubule walls and tips could potentially contribute to the pronounced variability in observable behaviors. This leads to the concern that interventions such as varying ATP concentration or introducing neck linker mutations might indirectly influence lateral movements by modulating the number of engaged motors. Addressing this concern directly entails two approaches: firstly, varying motor concentration in the original assays utilizing wild-type motors and 1 mM ATP; secondly, visualizing motor density through employment of GFP-tagged motors on overlapping microtubules or at least on immobilized unbundled microtubules on coverslips.

The outcome of these supplementary assays will serve to either dispel the "affinity" concern by demonstrating that motor density in isolation cannot account for the observed motility changes,

or potentially prompt an adjusted discussion that duly acknowledges this multifaceted complexity.

Response: We thank the reviewer for the suggestion and now performed additional experiments with various KIF11 concentrations (2, 5 and 50 nM; standard concentration in experiments 10 nM). Almost none of the parameters differed significantly. Hence, we can rule out that motor density affects the motility parameters of cargo microtubules. **We added the new Appendix Fig. 4 and changed the text accordingly.**

For example:

Current authors conclusion for ATP experiments: " This indicates, that the slower a cargo microtubule moves in the longitudinal direction, the more likely it moves sideways in the axial direction."

Possible explanation 2. If fewer motors are present at lower ATP, could this explain slower forward and sideways motility? The fact that the sidestepping probability is increased at lower speeds is not disputed but the cause may be different than suggested

Response: Our new data (**new Appendix Fig. 4**) does not indicate a dependence of the sliding velocity on the motor density. Thus, even if at low ATP concentration fewer motors were present, we would not expect a decrease in the velocity due to motor number.

Possible explanation 3: if more motor molecules accumulate at the microtubule tip during faster motility (or for different mutants), could this impede rotational stepping, thereby lowering its apparent probability?

Response: In our experiments, we did not observe a pronounced accumulation of motors at microtubule tips. Taken together with our newly added and discussed data (**new Appendix Fig. 4**) we do therefore not believe that the scenario described by the reviewer is at play.

Minor comments

1. Please describe the KIF11 construct used in this work early in the results section.

Response: We improved the description of the construct early on in the results section and provide a full description in the Materials and Methods.

2. The units for stepping probability are not intuitive. Please consider using fractions of 1 and clearly describe what this probability reflects

Response: We changed the parameter from "effective sidestepping probability" to "sidestepping ratio", which should be more intuitive. It is calculated the same way, but not multiplied by 100 and reflects the ratio of sideways to forward motion as described in the text.

3. Discussion

"Perturbing the neck linker length altered the motility parameters, indicating the KIF11 has evolved for slow and robust motility" Please modify. This study does not address evolution of KIF11 or robustness of its motility. Perhaps the case can be made that the linker is optimized for certain features, but the statement would benefit from evidence-based specificity

Response: We removed this statement.

(Also in abstract) "We conclude that the helical motion of microtubules driven by KIF11 allows for flexible, context-dependent filament organization and torque regulation in the mitotic spindle." The fact that the helicity has different polarity in cells and in vitro is very intriguing but it hardly allows to conclude that KIF11 regulates the torque or it is involved in "flexible, context-dependent filament organization"

Response: We changed the statement by replacing 'regulates' with 'contributes to'.

4. Introduction:

Page 3. "Recently, high resolution imaging of HeLa and RPE1 cells revealed, that the spindle is twisted into a chiral structure." Please improve writing, this statement contradicts literature cited in this paper, including the sentence in the same paragraph: "In contrast to HeLa cells, spindles of RPE1 cells did not exhibit helicity ..."

Response: We re-wrote this part to stress that spindles in HeLa cells are strongly twisted, but spindles in RPE1 are weakly twisted or did not show helicity [Novak et al. 2018, Trupinić et al. 22, Neahring et al. 21].

Page 3. "This suggests, that KIF11 has evolved as a slowly sliding, fast rotating motor ..". Please modify. This work has no bearing on KIF11 motor evolution.

Response: We changed the statement by replacing 'has evolved' by 'constitutes'.

Also, the rates of sliding and rotation are provided in this work in different units (page 4). Please supplement this or similar sentence with numerical results using the same units (perhaps steps per sec)

Response: Forward sliding velocities are given as nm/s and angular velocities are given as rad/s. If we were talking about the properties of a single motor molecules, it would indeed make sense to relate them to each other by putting them into "steps per sec". However, in our case we quantify the movement of one microtubule along/around another. We did however pick up the idea of the reviewer in the (now called) "stepping ratio" calculated as the ratio of sideways movement (in protofilament steps, i.e., in units of $2\pi/14$) to forward displacement (in steps of 8 nm).

5. Figure 3. Is directional velocity zero for all microtubules? If not, would it be possible to provide actual data?

Response: We consider cargo microtubules as "sideways only" when their forward velocity is 20% or less than the mean velocity of "forward and sideways" cargo microtubules. We added this now more clearly into the Materials and Methods. The actual velocity data is shown in Fig. 4B in black.

Figure 6 B,C is difficult to follow and it does not add much to discussion because this aspect seems very speculative and other features of the system may be at play. A cartoon or table that summarizes differences/similarities in sidestepping of different motors would enhance this last figure, but this is just a suggestion for authors.

Response: We agree. We revised Fig. 6.

Referee #4:

Recently, kinesin-5 has been implicated in the generation of microtubule twisting at the mitotic spindle microtubules. Meißner et al. study the helical motion of the cargo microtubule driven by tetrameric human kinesin-5 KIF11, which is an important player in prometaphase, along the freely suspended, fixed microtubule in vitro. By applying the experimental setups from their previous work on the kinesin-14 motor proteins, the authors found that tetrameric plus-end-directed kinesin-5, KIF11, slides the cargo microtubules around the fixed microtubule in a right-handed helical motion. The pitch of the helical motion depends on the ATP concentration, indicating changing the ratio of forward step to side step. Furthermore, this is a novel finding that extension of the KIF11 controls the forward velocity of antiparallel cargo microtubules. Overall, this study led to important new knowledge about the motility and regulation of torque component of KIF11 in MT-MT sliding. I support the publication of this work with several revisions.

Response: We thank the reviewer for the positive assessment and appreciation of our work as well for the valuable comments. Please see for our specific responses below.

My comments are as follows:

1) The authors state in the abstract: "~and found that the motor caused right-handed rotation of anti-parallel microtubules around each other." However, they did not observe the rotational motion of the cargo MT itself along its longitudinal axis. It is better to delete the description of the direction of rotation of the cargo microtubule along its long axis.

Response: We are sorry for the mistake and corrected 'rotation' with 'helical motion'.

In addition, in Fig 6A, authors show the direction of the microtubule rotation in the microtubule cross-section as a purple arrow, and the direction of rotation of kinesin as it moves around the microtubule surface as a gray arrow. These arrows need to be removed from the figure or the results need to be provided for the rotational direction of the sliding cargo MT itself along its long axis, driven by cross-linking KIF11 molecules.

Response: We agree that we did not directly observe the motion indicated by the purple and gray arrows. **We now stress that these motions are only inferred from our data by rendering these arrows as dashed lines.**

2) My main concern is that in Figure 3D, the helix direction appears to be left-handed, contrary to Figure 1C. Even considering the direction of the axis indicating light intensity and sideways distance, it appears to be left-handed.

Response: In our general analysis procedure, the orientation (direction of plus end) of the fixed microtubule is assigned and then the sideways distance is calculated. For forward and sideways moving cargo microtubules, this orientation of the fixed microtubule is easily determined by the sliding direction of the cargo microtubule. For sideways-only moving cargo microtubules,

however, the orientation of the fixed microtubule was usually not assigned, because it would have required unambiguous polarity-labeling in all events.

In fact, in the example event of Fig. 3D, the direction of the fixed microtubule was unclear. However, we confirmed a right-handed orbiting of sideways-only microtubules in multiple other events, where we lowered the focal plane. Thus, we infer that the orbiting of all sideways-only microtubules is right-handed (when viewed from the minus end of the cargo microtubule). **To avoid confusion, we changed the orientation of the fixed microtubule, resulting in an inverted sign of the sideways distance and a right-handed helix. Moreover, we refined our description of this strategy in the Materials and Methods.**

Although the parallel cargo microtubules are stated to orbit without forward motion, the cargo microtubule shown in the associated movies (S2 and S4) clearly appears to move forward along a fixed microtubule, i.e., helical motion. Movie 4 also shows that the cargo microtubule clearly moves forward along a fixed microtubule at several nm/s toward the minus end of the microtubule.

Response: Yes, the cargo microtubule is moving forward – but very slowly. As described in the main text, we don't consider velocities < 5 nm/s as significant forward motion and thus group these cargo microtubules into the category of "sideways only" (as comparison, all "forward and sideways" microtubules were sliding with > 20 nm/s). The two populations of the "sideways only" and "forward and sideways" microtubules are clearly separated from each other (compare black and purple data in Fig. 4, at 1 mM ATP).

Perhaps the plot shown in the figure 3D is for a parallel cargo microtubule moving (slowly) toward the minus end of a fixed microtubule, in which case it seems to have the same helical motion as a minus-end-directed kinesin-14 in a right-handed helical motion. If the authors intend to claim right-handed orbiting (counterclockwise rotation without forward motion) as seen from the minus end of the microtubule, a plot and its movie of the case with almost no forward motion, equivalent to Figure 3D, is needed. Also, it would be better to show the plot and its movie of the slow movement of a parallel cargo microtubule toward the plus end of a fixed microtubule as supplementary information.

Response: Please see two paragraphs above (our strategy of lowering the focal plane) for our reasoning of the handedness during the orbiting of parallel microtubules. Because the handedness and polarity were thus determined in two independent experiments, we cannot present an event with both data in it. However, we believe that the above described strategy (**now further detailed on in the Materials and Methods**) answers the question raised by the reviewer.

3) It is better to show which end of the fixed microtubule is the + or - end in Movie 2

Response: As described in the main text, we classified all events with sliding velocities lower than 5 nm/s as parallel events after confirming this correlation with polarity-marked microtubules (see p. 7: 'We detected five events of helically moving cargo microtubules with four of them in an anti-parallel and one in a parallel orientation (Appendix Fig. 6E, Supplementary Movie 3). In contrast, from 18 sideways-only cargo microtubules all of them were in a parallel configuration (Appendix Fig. 6F, Supplementary Movie 4), confirming our hypothesis.'). For the event shown in Movie S2 the polarity of the fixed microtubule has not been explicitly determined but rather

the velocity argument has been applied. An example of an orbiting of parallel microtubule with clear polarity marking is shown in Movie S4 (and see also Appendix Fig. 6F).

4) There are several issues regarding the description of the number of samples in the experiments. For instance, in Figure 2, the number of ATP concentration-dependent assays is not provided.

Response: We added the sample number of the plots in Fig. 2A-D in the figure legend and text as well as in the legend of Fig. 5.

Additionally, there is a discrepancy in the number of "sideways-only polarity-labeled cargo microtubules", which are stated as 18 and 9 in the same section. Can you clarify the difference between these two numbers?

Response: We could determine the polarity of fixed and cargo microtubules of 18 sideways-only events. To determine the helicity, we performed control experiments with a lowered focal plane. In this way the change of the signal intensity was in a defined, unidirectional direction. Here three cargo microtubules without forward motion (no simultaneous polarity labeling was performed) orbited in a right-handed manner. We mistakenly stated 9 microtubules – this was the number of microtubules with a clear change in signal intensity. However, we could not unambiguously determine the position of the focal plane relative to the microtubules and therefore the handedness. **We now clarified this in the text.**

Also, while the representation of "motor forward" microtubules are referred to as both 47% and 306 events, for a total of 651 events, the representation of "immobile in forward" microtubules are presented as 53% and 341 events, for a total of 643 events. Can you explain the reason for the difference in numbers? For clarity, please provide a clear breakdown of the number of samples in the experiments. This will help to understand the details.

Response: The percentages were rounded, which led to this discrepancy. **We now describe more precisely (one more significant digit) that 47.3% of cargo microtubules were mobile, 52.7% immobile, adding up to 647 events and provide a breakdown of data sets in Appendix Table 1.**

Dr. Stefan Diez
TU Dresden
B CUBE
Tatzberg 41
Saxony 01307
Germany

22nd Jan 2024

Re: EMBOJ-2023-115116R

Kinesin-5 drives the helical motion of anti-parallel and parallel microtubules around each other

Dear Stefan,

Thank you again for submitting your revised manuscript to The EMBO Journal. Three of the original referees have now reviewed it once more, and as you will see from their comments below, are all fully satisfied with your revisions. We are therefore happy to accept the study for publication, following incorporation of several editorial points as follows:

1) Issues with the main manuscript text:

- Main Figure legends have to be included here, after the reference section (while main figures themselves should be uploaded separately as figure files without included legends).
- A reference to Figure 3D appears to be missing and should be added
- When referencing the movies, please remove the word "supplementary" - should just be "Movie EV1/2/..."
- Please correct the header of the competing interest section to "Disclosure and competing interests statement"
- As we are switching from a free-text author contribution statement towards a more formal statement based on Contributor Role Taxonomy (CRediT) terms, please remove the present Author Contribution section and instead specify each author's contribution(s) directly in the Author Information page of our submission system during upload of the final manuscript. See <https://casrai.org/credit/> for more information.
- Please correct the reference list according to EMBO Journal style: remove hyperlinks or DOI information (accept for preprints or prepublications), always provide proper journal names (e.g. case of Goulet & Moores?) and full volume and page number/locator information (missing e.g. for Murayama et al, or for several eLife citations).

2) Issues with the Appendix PDF:

- Please call/list the included M&M and Reference sections as "Appendix Material and Methods" and "Appendix References", both in the table of contents and as their header.
- Please remove the Data Availability section in the Appendix, it is enough to have it in the main text
- Please include the two Appendix Tables (and their respective page numbers) in the table of contents.
- On page 2, please rename the header from "Supplementary Figures" to "Appendix Figures"

3) Synopsis material:

- Please provide suggestions for a short 'blurb' text prefacing and summing up the study in two sentences (max. 250 characters), followed by 3-5 one-sentence 'bullet points' with brief factual statements about key results of the paper; they will form the basis of an editor-written 'Synopsis' accompanying the online version of the article (see new articles on our journal website for some recent examples). Please also provide a simple synopsis image, which can be used as a "visual title" for the synopsis section of your paper (maybe based on the schematics in Fig 6?). The image should be in PNG or JPG format with the modest dimensions of 550 x 300-600 pixels (width x height).

4) Source Data files:

- Please combine the source data files for the APPENDIX FIGURES in one single ZIP archive before re-uploading. The main figure source data are already correctly uploaded as one file per figure, but Appendix SD needs to be combined.

I am therefore returning the manuscript to you for a final round of minor revision, to allow you to make these adjustments and upload all modified files. Once we will have received them, we should be ready to swiftly proceed with formal acceptance and production of the manuscript!

With kind regards,

Hartmut

Hartmut Vodermaier, PhD

*** PLEASE NOTE: All revised manuscripts are subject to initial checks for completeness and adherence to our formatting guidelines. Revisions may be returned to the authors and delayed in their editorial re-evaluation if they fail to comply to the following requirements (see also our Guide to Authors for further information):

9) Digital image enhancement is acceptable practice, as long as it accurately represents the original data and conforms to community standards. If a figure has been subjected to significant electronic manipulation, this must be clearly noted in the figure legend and/or the 'Materials and Methods' section. The editors reserve the right to request original versions of figures and the original images that were used to assemble the figure. Finally, we generally encourage uploading of numerical as well as gel/blot image source data; for details see: embopress.org/page/journal/14602075/authorguide#sourcedata

At EMBO Press, we ask authors to provide source data for the main manuscript figures. Our source data coordinator will contact you to discuss which figure panels we would need source data for and will also provide you with helpful tips on how to upload and organize the files.

Further information is available in our Guide For Authors:

In the interest of ensuring the conceptual advance provided by the work, we recommend submitting a revision within 3 months (21st Apr 2024). Please discuss the revision progress ahead of this time with the editor if you require more time to complete the revisions. Use the link below to submit your revision:

Link Not Available

Referee #1:

The authors have nicely improved what was already quite a well-presented manuscript at first submission and satisfactorily addressed all my concerns, and as far as I can see also the concerns of the other reviewers. The Discussion is maybe still a bit lengthy, but that's up to the authors.

Referee #3:

The authors have addressed all my prior criticism and I fully support accepting this manuscript

Referee #4:

I am satisfied with the answers to my suggestions/comments and I have no further remarks. I think the revisions in response to all the comments have significantly improved the manuscript. In my opinion, the revised manuscript is suitable for publication.

Dr. Stefan Diez
TU Dresden
B CUBE
Tatzberg 41
Saxony 01307
Germany

29th Jan 2024

Re: EMBOJ-2023-115116R1
Kinesin-5 drives the helical motion of anti-parallel and parallel microtubules around each other

Dear Stefan,

Thank you for submitting your final revised manuscript for our consideration. I am pleased to inform you that we have now accepted it for publication in The EMBO Journal.

With kind regards,

Hartmut
